# DMFlow: Disordered Materials Generation by Flow Matching

## Abstract

The design of materials with tailored properties is crucial for technological progress. However, most deep generative models focus exclusively on perfectly ordered crystals, neglecting the important class of disordered materials. To address this gap, we introduce DMFlow, a generative framework specifically designed for disordered crystals. Our approach introduces a unified representation for ordered, Substitutionally Disordered (SD), and Positionally Disordered (PD) crystals, and employs a flow matching model to jointly generate all structural components. A key innovation is a Riemannian flow matching framework with spherical reparameterization, which ensures physically valid disorder weights on the probability simplex. The vector field is learned by a novel Graph Neural Network (GNN) that incorporates physical symmetries and a specialized message-passing scheme. Finally, a two-stage discretization procedure converts the continuous weights into multi-hot atomic assignments. To support research in this area, we release a benchmark containing SD, PD, and mixed structures curated from the Crystallography Open Database. Experiments on Crystal Structure Prediction (CSP) and De Novo Generation (DNG) tasks demonstrate that DMFlow significantly outperforms state-of-the-art baselines adapted from ordered crystal generation. We hope our work provides a foundation for the AI-driven discovery of disordered materials.

## 1 Introduction

The search for novel materials with tailored properties is a fundamental goal in technological advancement. In recent years, deep generative models (Xie et al., 2022; Jiao et al., 2023; Miller et al., 2024; Zeni et al., 2025; Wu et al., 2025b) have emerged as powerful tools to accelerate this process. Compared with traditional search algorithms (Laks et al., 1992; van de Walle et al., 2013), which are computationally expensive, deep generative models can not only discover novel and stable crystal structures with high efficiency but also integrate fundamental physical principles, such as the periodicity invariance that governs crystalline materials. Despite these advances, most existing approaches have focused almost exclusively on perfectly ordered crystals, overlooking the vast and technologically important class of materials that inherently exhibit disorder.

Disordered crystals, whose unique and valuable properties arise from their intrinsic lack of perfect periodicity, are foundation forms in high-entropy alloys, solid-state electrolytes, and superconductors (Seo et al., 2014; Maiti & Steurer, 2016; Botros & Janek, 2022). As illustrated in Fig. 1, two prevalent types of disorder are Substitutional Disorder (SD) and Positional Disorder (PD). Specifically, SD refers to chemical disorder, where distinct atomic species can probabilistically occupy the same site of different unit cells (Fig. 1(b)). This behavior is described by a substitutional vector, with entries indicating the probability of each species at that site. PD arises from structural deviations where atoms are deviated from their ideal positions (Fig. 1(c)). Such deviations are described by a positional vector, whose entries denote the probability of an atom residing at different possible lo-

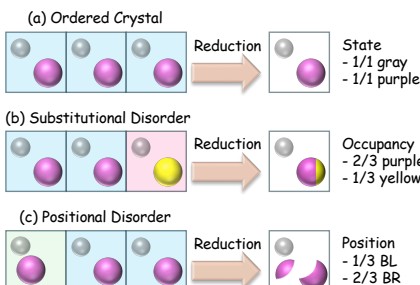

Figure 1: Ordered vs. Disordered crystals. Arrows show supercell reduction; BL/BR mark bottom-left/right.

cation. For consistency, both SD and PD vectors are termed *disorder weights*, with their entries constrained to sum to one, forming a probability distribution.

The probabilistic nature of disordered crystals, however, introduces significant challenges that hinder a direct application of existing generative models. Current models for crystal generation are fundamentally built on the assumption of perfect periodicity, where atomic types and their precise coordinates are deterministically repeated throughout 3D space. Consequently, they lack a coherent and efficient representation for the probabilistic site occupancies for SD systems and position derivations for PD systems. A further difficulty lies in generating the disorder weights themselves: these vectors must satisfy the simplex constraint to constitute valid probability distributions (Nelder & Mead, 1965), whereas existing generative models typically operate in Euclidean space for continuous variables or a one-hot vector space for discrete variables. Finally, and likely due to the aforementioned difficulties, there is no established public benchmark for generative modeling of disordered crystals.

To overcome these challenges, we introduce *DMFlow*, the first deep generative framework designed for disordered crystal generation. At the core of our approach is a unified representation that seamlessly accommodates ordered, SD, and PD crystals within a single framework, which not only simplifies the generation task but also enables mixed-data training. Building on this representation, DMFlow adopts a holistic generative model based on flow matching (Lipman et al., 2023), which jointly generates the lattice, fractional coordinates, disorder weights, and PD displacement coordinates, thereby capturing the intricate inter-dependencies among all structural components. Particularly, to ensure physically valid disorder weights under the simplex constraint, DMFlow uses a Riemannian flow matching framework with spherical reparameterization. This avoids the numerical instabilities of a direct simplex implementation, ensuring a stable and well-conditioned flow matching process.

The vector field of the generation path is learned by a novel Graph Neural Network (GNN). This architecture is designed to ensure physical realism by enforcing key symmetries like periodic invariance, supporting continuous disorder weights beyond conventional one-hot vectors, and using a message-passing scheme tailored to the complex, multi-position interactions in PD systems. After generation, the continuous disorder weights must be converted into multi-hot vectors, reflecting discrete atomic assignments. To achieve this, we introduce a robust, two-stage discretization procedure comprising ordered site identification and ensemble voting for disordered sites.

Finally, to facilitate the research in this area, we construct and release the first public benchmark containing structures with SD, PD, and SPD (the mixture of SD and PD) curated from the Crystallography Open Database (COD) (Gražulis et al., 2009). We investigate two key tasks on this new bechmark: Crystal Structure Prediction (CSP), which predicts a structure for a given composition, and *De Novo* Generation (DNG), which discovers novel materials from scratch.

In summary, our contributions are threefold: **(1)** We propose DMFlow, a unified generative model for representing and generating both SD and PD crystals. **(2)** We construct the first benchmark dedicated to evaluating generative models on disordered materials. **(3)** We implement and evaluate DMFlow on this benchmark, demonstrating its promising performance for both CSP and DNG tasks.

## 2 RELATED WORK

**Deep Learning for Disordered Crystals.** The study of disordered materials has long relied on computationally intensive simulation and search algorithms. Recently, deep learning on disordered systems has so far focused primarily on property prediction. For instance, GNNs have been trained on statistical ensembles of local atomic environments to predict properties of high-entropy alloys (Zhang et al., 2024), while Chen et al. (2024) explicitly encodes site occupancies to estimate the critical temperature of disordered superconductors. Although effective for prediction, these methods remain fundamentally discriminative rather than generative. A recent step toward generative modeling involved training a Variational Autoencoder (VAE) (Kingma & Welling, 2013) on Wyckoff positions to sample disordered candidates (Petersen et al., 2025), but its rigid representation and inability to capture PD significantly limit its applicability. In comparison, DMFlow introduces a feasible generative framework, enabling true generative discovery rather than mere prediction.

**Generative Models for Ordered Crystals.** In parallel, advances in deep generative models have revolutionized materials discovery, particularly for ordered crystals. VAEs adopt a two-stage strat-

egy, first sampling lattices and then placing atoms (Xie et al., 2022; Luo et al., 2024a). Diffusion models unify these steps by jointly denoising lattices, coordinates, and compositions (Jiao et al., 2023; 2024; Lin et al., 2024; Zeni et al., 2025; Wu et al., 2025b). Flow-based models learn invertible mappings on the Riemannian manifolds of crystal structures, allowing exact likelihood estimation (Miller et al., 2024; Luo et al., 2024b). Most recently, Bayesian Flow Networks (BFNs) have emerged as a competitive framework, generating crystals through iterative Bayesian inference that continuously refines structural parameters under periodic and symmetric constraints (Wu et al., 2025a; Ruple et al., 2025). Despite their success, these methods are fundamentally constrained by the assumption of perfect periodicity and lack the mechanisms needed to represent disorder. DMFlow directly addresses this gap by extending generative modeling beyond ordered crystals, introducing a unified representation and architecture capable of capturing both SD and PD.

## 3 BACKGROUND

Flow matching learns a time-dependent velocity field $v_t(x)$ that transports samples from a prior $p_0(x)$ to a target distribution $p_1(x)$. Conditional Flow Matching (CFM) (Lipman et al., 2023) circumvents the intractability of the marginal vector field $u_t(x)$ by introducing conditional paths $p_t(x|x_1)$, yielding $u_t(x) = \mathbb{E}_{x_1 \sim p_1}[u_t(x|x_1)]$, so that training reduces to regressing $v_\theta(x_t, t)$ onto the tractable conditional field. For physical systems such as crystals, distributions reside on a non-Euclidean manifold $\mathcal{M}$. Riemannian Flow Matching (Chen & Lipman, 2024) defines $v_t(x)$ in the tangent space $\mathcal{T}_x\mathcal{M}$, with trajectories parameterized by geodesics via $x_t = \exp_{x_0}(t \cdot \log_{x_0}(x_1))$, where $\exp$ and $\log$ are the exponential and logarithm maps linking $\mathcal{M}$ and $\mathcal{T}_x\mathcal{M}$. This ensures generated trajectories respect the underlying geometry. More details are given in Appendix B.1.

## 4 METHODOLOGY

### 4.1 A UNIFIED REPRESENTATION FOR DISORDERED CRYSTALS

A key challenge in generating disordered materials is developing a representation that can uniformly handle different types of disorder. Our approach is to represent crystals from the perspective of each crystallographic site, starting from the perfectly ordered case and generalizing to SD and PD cases. This provides an intuitive and comprehensive description of how disorder manifests.

For **ordered** case, each site $i$ is deterministically occupied by a single atom, described by a pair $(\boldsymbol{a}_i, \boldsymbol{f}_i)$, where $\boldsymbol{a}_i \in \{0,1\}^D$ is a one-hot vector for the atomic type (with $D$ being the total number of atom types considered) and $\boldsymbol{f}_i \in [0,1)^3$ is the fractional coordinate. To generalize this to disordered cases, we introduce probabilistic representations for atomic types and positions.

For **SD**, we replace the one-hot vector $\boldsymbol{a}_i$ with an occupancy vector $\boldsymbol{s}_i = [s_{i,0}, \ldots, s_{i,D-1}]^\top \in \Delta^{D-1}$. This vector $\boldsymbol{s}_i$ resides on a $(D-1)$-simplex, satisfying $\sum_{k=0}^{D-1} s_{i,k} = 1$ and $s_{i,k} \geq 0$. Each component $s_{i,k}$ represents the probability of assigning an atom of element type $k$ at site $i$. The ordered case is a special instance when $\boldsymbol{s}_i$ is reduced to a one-hot vector.

Regarding **PD**, we introduce two new components to describe atoms deviating from their primary positions. Our formulation specifically addresses the binary case, where one site can occupy no more than two possible positions. This choice is motivated by our dataset from COD (Gražulis et al., 2009), in which binary PD is prevalent. Therefore, we define a positional vector $\boldsymbol{w}_i = [w_{i,0}, w_{i,1}]^\top \in \Delta^1$ to represent the occupancy probabilities of the two possible locations, alongside an additional fractional coordinate $\boldsymbol{f}_i' \in [0,1)^3$ for the secondary location, while the primary position remains $\boldsymbol{f}_i$. Accordingly, $w_{i,0}$ and $w_{i,1}$ denote the probabilities of the atom appearing at positions $\boldsymbol{f}_i$ and $\boldsymbol{f}_i'$, respectively. A structure without PD becomes the case where $\boldsymbol{w}_i = [1,0]^\top$ and $\boldsymbol{f}_i'$ is a zero vector. While cases involving higher-order PD (more than two positions) are less common in this dataset, our framework can be naturally extended to handle them (see Appendix E).

Combining the above definitions, we can present a **Unified Representation**. Any crystallographic site $i$ in a potentially disordered material is described by the tuple $(\boldsymbol{s}_i, \boldsymbol{f}_i, \boldsymbol{w}_i, \boldsymbol{f}_i')$. The complete disordered crystal structure $\mathcal{C}$ with $N$ sites is then fully defined by:

$$\mathcal{C} := (\boldsymbol{L}, \{(\boldsymbol{s}_i, \boldsymbol{f}_i, \boldsymbol{w}_i, \boldsymbol{f}_i')\}_{i=1}^N) = (\boldsymbol{L}, \boldsymbol{S}, \boldsymbol{F}, \boldsymbol{W}, \boldsymbol{F}'), \tag{1}$$

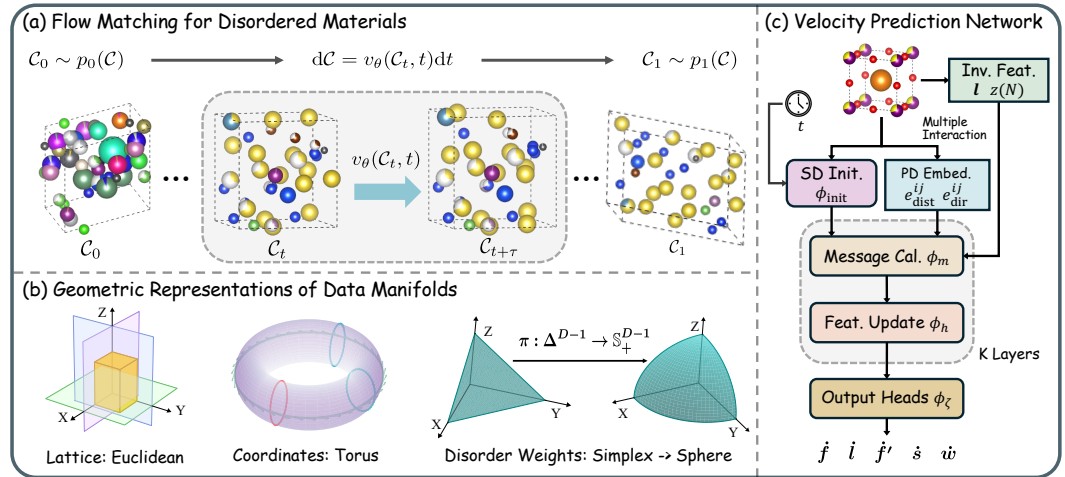

Figure 2: Overall framework of DMFlow. (a) The flow matching process, which transports samples from a prior distribution to the target data distribution; **balls with a white segment indicate empty occupancy, corresponding to PD**. (b) Geometric representations of the distinct manifolds underlying disordered crystals, on which our conditional flow matching operates. (c) The architecture of our GNN backbone used for velocity prediction.

where $\boldsymbol{L} \in \mathbb{R}^{3\times3}$ denotes the lattice matrix, and $\boldsymbol{S}, \boldsymbol{F}, \boldsymbol{W}, \boldsymbol{F}'$ are the corresponding matrix representations of the site-level features. This flexible and powerful representation can seamlessly describe ordered, SD and PD crystals, or materials exhibiting both types of disorder simultaneously.

## 4.2 GENERATIVE PATHWAY FOR DISORDERED CRYSTALS

We design our generative model by applying the CFM framework to the aforementioned unified representation, as illustrated in Fig. 2. A crucial aspect of this design is to define appropriate manifold, probability path, and vector field for each quantity describing the structure $\mathcal{C} := (\boldsymbol{L}, \boldsymbol{S}, \boldsymbol{F}, \boldsymbol{W}, \boldsymbol{F}')$.

**Flow Matching on Lattice and Fractional Coordinates.** Following FlowMM (Miller et al., 2024), we adopt its flow matching setup for both lattice parameters and fractional coordinates. For the lattice, let $\tilde{l}$ denote the unconstrained representation of the Niggli-reduced parameters. Given interpolated samples $\tilde{l}_t$ at interpolation time $t \in [0, 1]$, the training loss is

$$\mathcal{L}_{\tilde{l}} = \mathbb{E}_{t\in\mathcal{U}(0,1),p(\tilde{l}_1),p(\tilde{l}_t|\tilde{l}_1)} \left[ \frac{1}{6} \|v_{\theta,\tilde{l}}(\tilde{l}_t, t) - (\tilde{l}_1 - \tilde{l}_0)\|_2^2 \right]. \tag{2}$$

For the fractional coordinates $\boldsymbol{F}$ of $N$ atoms, with prior $\boldsymbol{F}_0$ and data $\boldsymbol{F}_1$, we define normalized displacements $\hat{s}(\boldsymbol{F}_0, \boldsymbol{F}_1)$ and interpolated states $\boldsymbol{F}_t$. The corresponding loss is

$$\mathcal{L}_{\boldsymbol{F}} = \mathbb{E}_{t\in\mathcal{U}(0,1),p(\boldsymbol{F}_1),p(\boldsymbol{F}_t|\boldsymbol{F}_1)} \left[ \frac{1}{3N} \|v_{\theta,\boldsymbol{F}}(\boldsymbol{F}_t, t) - \hat{s}(\boldsymbol{F}_0, \boldsymbol{F}_1)\|_2^2 \right]. \tag{3}$$

For PD sites, the secondary coordinates $\boldsymbol{F}'$ are modeled identically with the same objective. All details are provided in Appendix B.2.

**Flow Matching on Disorder Weights.** For simplicity, we denote the disorder weights as a categorical distribution $\boldsymbol{\mu} = (\mu_1, \ldots, \mu_D)$, corresponding to substitutional vector $\boldsymbol{s}_i \in \Delta^{D-1}$ or positional vector $\boldsymbol{w}_i \in \Delta^1$. Notably, $\boldsymbol{\mu}$ is constrained to lie on a simplex, where a natural Riemannian geometry is given by the *Fisher-Rao metric* (Atkinson & Mitchell, 1981; Rao, 1992):

$$g_{\boldsymbol{\mu}}^{\text{FR}}(\boldsymbol{u}, \boldsymbol{v}) = \sum_{k=1}^{D} \frac{u_k v_k}{\mu_k}, \tag{4}$$

for tangent vectors $\boldsymbol{u}, \boldsymbol{v} \in T_{\boldsymbol{\mu}}\Delta^{D-1}$ satisfying $\sum_k u_k = \sum_k v_k = 0$. And the Fisher-Rao geodesic distance of two categorical distribution $\boldsymbol{\mu}, \boldsymbol{\nu}$ follows the closed form as:

$$d_{\text{FR}}(\boldsymbol{\mu}, \boldsymbol{\nu}) = \inf_{\gamma} \int_0^1 \sqrt{g_{\gamma(t)}^{\text{FR}}(\gamma'(t), \gamma'(t))} dt = 2\arccos\left( \sum_{k=1}^{D} \sqrt{\mu_k \nu_k} \right), \tag{5}$$

where $\gamma : [0, 1] \to \Delta^{D-1}$ is a smooth curve on the simplex with $\gamma(0) = \boldsymbol{\mu}, \gamma(1) = \boldsymbol{\nu}$. However, Eq. (4) diverges at the boundaries where $\mu_k \to 0$, which makes direct flow matching on $\Delta^{D-1}$ numerically unstable. To obtain a well-conditioned geometry, we adopt the spherical reparameterization (Cheng et al., 2024). We define the mapping $\pi(\boldsymbol{\mu}) = \sqrt{\boldsymbol{\mu}} \in \mathbb{S}_+^{D-1}$, which sends $\boldsymbol{\mu}$ to the positive orthant of the unit sphere. Under this transformation, the spherical distance serves as the geometric realization of the Fisher–Rao distance:

$$d_{\mathbb{S}}(\pi(\boldsymbol{\mu}), \pi(\boldsymbol{\nu})) = \arccos(\textstyle\sum_{k=1}^{D} \sqrt{\mu_k \nu_k}) = \tfrac{1}{2} d_{\text{FR}}(\boldsymbol{\mu}, \boldsymbol{\nu}). \tag{6}$$

Thus, flow matching on the simplex reduces to Riemannian flow matching on the sphere, where the closed-form expressions for exponential and logarithmic maps are available:

$$\exp_{\pi(\boldsymbol{\mu})} \boldsymbol{v} = \cos \|\boldsymbol{v}\| \pi(\boldsymbol{\mu}) + \operatorname{sinc}\|\boldsymbol{v}\| \boldsymbol{v}, \tag{7}$$

$$\log_{\pi(\boldsymbol{\mu})} \pi(\boldsymbol{\nu}) = \operatorname{sinc}^{-1} d_{\mathbb{S}}(\pi(\boldsymbol{\mu}), \pi(\boldsymbol{\nu}))(\pi(\boldsymbol{\nu}) - \langle \boldsymbol{\mu}, \boldsymbol{\nu} \rangle^{\frac{1}{2}} \pi(\boldsymbol{\mu})), \tag{8}$$

where $\operatorname{sinc}(x) = \lim_{x' \to x} \frac{\sin(x')}{x'}$. Formally, given a prior sample $\boldsymbol{\mu}_0$ and a data sample $\boldsymbol{\mu}_1$, the geodesic interpolation path is:

$$\boldsymbol{\mu}_t = \pi^{-1}(\exp_{\pi(\boldsymbol{\mu}_0)}(t \cdot \log_{\pi(\boldsymbol{\mu}_0)}(\pi(\boldsymbol{\mu}_1)))), \quad t \in [0, 1], \tag{9}$$

where $\pi^{-1}$ denotes the elementwise square operation.

For the prior distribution we choose the uniform distribution on the simplex, which corresponds to the normalized uniform distribution on $\mathbb{S}_+^{D-1}$ after the mapping $\pi$. The training loss is defined as

$$\mathcal{L}_{\boldsymbol{\mu}} = \mathbb{E}_{t \sim \mathcal{U}(0,1),\, p(\boldsymbol{\mu}_1),\, p(\boldsymbol{\mu}_t|\boldsymbol{\mu}_1)} \left[ \frac{1}{ND} \left\| v_{\theta,\boldsymbol{\mu}}(\boldsymbol{\mu}_t, t) - \log_{\pi(\boldsymbol{\mu}_0)}(\pi(\boldsymbol{\mu}_1)) \right\|_2^2 \right], \tag{10}$$

where $\boldsymbol{\mu}$ can be instantiated as the substitutional vectors $\boldsymbol{s}_i$ or the positional vectors $\boldsymbol{w}_i$.

**Overall Training Objective.** The overall training objective is formulated as a weighted loss function that aggregates the targets from all fields. Detailedly, we have

$$\mathcal{L}_{\text{DMFlow}} = \lambda_{\tilde{l}} \mathcal{L}_{\tilde{l}} + \lambda_{\boldsymbol{F}} \mathcal{L}_{\boldsymbol{F}} + \lambda_{\boldsymbol{F}'} \mathcal{L}_{\boldsymbol{F}'} + \lambda_{\boldsymbol{S}} \mathcal{L}_{\boldsymbol{S}} + \lambda_{\boldsymbol{W}} \mathcal{L}_{\boldsymbol{W}}. \tag{11}$$

Specifically for CSP task where the occupancies are already given, we set $\boldsymbol{S}_t = \boldsymbol{S}_0, \boldsymbol{W}_t = \boldsymbol{W}_0$, and $\lambda_{\boldsymbol{S}} = \lambda_{\boldsymbol{W}} = 0$. Moreover, the loss $\mathcal{L}_{\boldsymbol{F}'}$ is computed only for PD sites.

## 4.3 Velocity Prediction Network

The velocity field $v_\theta$ is parameterized by a neural network that respects the physical symmetries of crystal structures. Prior works have successfully leveraged GNNs for this purpose (Jiao et al., 2023; Miller et al., 2024). However, these architectures are intrinsically restricted to perfectly ordered crystals, as they lack mechanisms to encode disorder weights, and model the complex interaction between different disordered sites.

To overcome these limitations, we introduce a novel GNN architecture (see Fig. 2(c)), which operates on both SD and PD sites. Specifically, each site $i$ is associated with a substitutional vector $\boldsymbol{s}_i$, fractional coordinates $\boldsymbol{f}_i$ and $\boldsymbol{f}_i'$, and the corresponding positional vector $\boldsymbol{w}_i = (w_{i,0}, w_{i,1})$ that quantify the occupancy of the two PD states. For clarity, we denote the two fractional coordinates $\boldsymbol{f}_i, \boldsymbol{f}_i'$ as $\boldsymbol{f}_{i,0}, \boldsymbol{f}_{i,1}$ below. The initial feature of site $i$ is obtained by projecting $\boldsymbol{s}_i$ and the timestep $t$:

$$\boldsymbol{h}_i^{(0)} = \varphi_{\text{init}}(\varphi_{\text{prob}}(\boldsymbol{s}_i), \varphi_{\text{time}}(t)), \tag{12}$$

where $\varphi_{\text{init}}, \varphi_{\text{prob}}, \varphi_{\text{time}}$ are three distinct MLPs. This design departs from conventional approaches, which typically rely on one-hot vectors for initialization.

Another key contribution of our backbone lies in the design of geometric edge embeddings. When constructing edge features for a pair of atoms $(i, j)$, we consider all combinations of their PD states, i.e., $(a, b) \in \{0, 1\} \times \{0, 1\}$, where $a$ indexes the state of atom $i$ and $b$ indexes the state of atom $j$. Each interaction is then weighted by the joint occupancy probability $w_{i,a} w_{j,b}$:

$$\boldsymbol{e}_{\text{dist}}^{ij} = \bigoplus_{a,b \in \{0,1\}} \left( w_{i,a} w_{j,b} \cdot \operatorname{SinusoidalEmb}\left( \log_{\boldsymbol{f}_{i,a}}(\boldsymbol{f}_{j,b}) \right) \right), \tag{13}$$

$$\boldsymbol{e}_{\text{dir}}^{ij} = \bigoplus_{a,b \in \{0,1\}} \left( w_{i,a} w_{j,b} \cdot \frac{\boldsymbol{M}(\boldsymbol{l})\,(\boldsymbol{f}_{j,b} - \boldsymbol{f}_{i,a})}{\|\boldsymbol{M}(\boldsymbol{l})\,(\boldsymbol{f}_{j,b} - \boldsymbol{f}_{i,a})\|} \right), \tag{14}$$

where $\bigoplus$ denotes concatenation, $e_{\text{dist}}^{ij}$ encodes the geodesic distance on the coordinate torus using a sinusoidal representation, while $e_{\text{dir}}^{ij}$ concatenates the four normalized directional vectors. The transformation $M(l) = L^{\top}L$, where $L$ is the lattice matrix parameterized by the lattice parameters $l$, ensures that the directional information faithfully reflects the underlying crystal geometry. Notably, the geometric edge features are not limited to binary PD. The underlying computation generalizes easily to higher-order cases, as detailed in Appendix E.

With the initial node (site) features and geometric embeddings established, the network performs $K$ layers of message passing and updates. At each layer $k$, a message $m_{ij}^{(k)}$ (Eq. (15)) is constructed based on the node features of sites $i$ and $j$, unconstrained lattice parameters $\tilde{l}$, the edge embeddings, and a learnable embedding of the total atom count, $z(N)$.

$$m_{ij}^{(k)} = \varphi_m\left(h_i^{(k-1)}, h_j^{(k-1)}, \tilde{l}, e_{\text{dist}}^{ij}, z(N), e_{\text{dir}}^{ij}\right), \quad (15)$$

$$m_i^{(k)} = \sum_{j \neq i} m_{ij}^{(k)}, \quad (16)$$

$$h_i^{(k)} = h_i^{(k-1)} + \varphi_h(h_i^{(k-1)}, m_i^{(k)}), \quad (17)$$

$$\dot{\zeta}_i = \varphi_\zeta(h_i^{(K)}), \quad \zeta \in \{f, f', s, w\}, \quad (18)$$

$$\dot{l} = \varphi_l(\text{Pool}(\{h_i^{(K)}\})). \quad (19)$$

After the message propagation (Eqs. (16) and (17)), the node features of the final layer are passed to a set of property-specific output heads to predict the velocity fields. The general formulation for node-level properties is shown in Eq. (18). For each property $\zeta$ (representing coordinates and disorder weights), a distinct MLP, denoted $\varphi_\zeta$, is used to predict its corresponding velocity. Concurrently, a graph-level feature, obtained by pooling the node embeddings, is used to predict the velocity for the lattice parameters (Eq. (19)).

## 4.4 DISCRETIZATION OF PROBABILISTIC REPRESENTATIONS

After generation, the model's probabilistic outputs must be converted into concrete atomic assignments. For each site $i$, these outputs consist of disorder weights: a substitutional vector $s_i' \in \Delta^{D-1}$ and a positional vector $w_i' \in \Delta^1$. Since the procedure for both cases is the same, we only describe discretization process for SD as an example: $s_i'$ is transformed into a *multi-hot vector* specifying which elements occupy site $i$. The main challenge lies in deciding how many elements to include (*i.e.*, the sparsity of the vector), which we address with a robust two-stage procedure as follows.

**Stage I: Detecting Ordered Sites.** The first stage identifies sites that are effectively ordered, meaning they are occupied by a single element type. We evaluate the sharpness of the predicted distribution by computing the ratio between the highest and the second-highest probability in $s_i'$. If this ratio exceeds a threshold, the site is deemed sufficiently certain to be ordered, and we assign it a one-hot vector corresponding to the most probable element.

**Stage II: Ensemble Voting for Disordered Sites.** Sites that do not meet the first-stage criterion are potentially treated as disordered, and their atomic assignments are inferred through an ensemble voting scheme (see Fig. 3). Specifically, we generate candidate multi-hot vectors using five complementary heuristics: a top-$k$ rule that selects the $k$ most probable elements, a fixed threshold rule that includes elements above a constant cutoff, a percentile-based rule that retains elements above a chosen quantile, an adaptive threshold rule that compares probabilities to a fraction of the maximum, and an entropy-based rule that decides between using *argmax* or adaptive top-$k$ depending on the

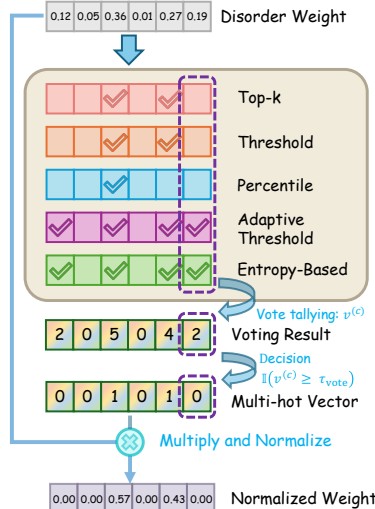

Figure 3: Ensemble voting.

normalized entropy of $s_i'$. The final assignment is obtained by majority voting across these candidates: an element is included in the multi-hot representation for site $i$ if its vote count reaches a minimum threshold. Notably, this procedure may still yield a one-hot vector when all heuristics consistently favor a single element, ensuring that ordered cases can naturally recovered as well. This ensemble strategy provides robust and physically consistent predictions while mitigating the bias of any single heuristic. Further details are provided in Appendix C.

## 5 EXPERIMENTS

**Benchmark Construction.** A well-curated benchmark is crucial for evaluating generative models of disordered crystals. As no such benchmark exists, we construct one from COD (Gražulis et al., 2009), parsing entries with disorder annotations. Given that the number of materials with only PD is relatively small, training a model exclusively on this subset is challenging. Therefore, we structure our benchmarks into two primary categories for evaluation: a set containing purely SD (COD-SD) and a comprehensive mixed set combining both SD and PD structures (COD-SPD). It is important to note that we only consider the binary PD cases, as explained in Section 4.1. We filter all structures to retain only those with 3 to 50 atoms per unit cell. Following the evaluation protocols in ordered crystal generation, we further partition our data by the number of atoms. Specifically, we create two subsets for structures with up to 20 atoms and 50 atoms. This results in the datasets COD-SD-20 and COD-SD-50, containing 5701 and 9096 structures respectively, as well as COD-SPD-20 and COD-SPD-50 with 6127 and 11746 structures. For each dataset, we use a standard 80%/10%/10% random split for training, validation, and testing. Appendix E presents higher-order PD datasets.

### 5.1 DISORDERED CRYSTAL STRUCTURE PREDICTION

**Baselines.** As our work introduces the first generative framework specifically designed for disordered materials, there is no direct baseline. To enable a rigorous comparison, we adapt three models originally developed for ordered crystal generation: **DiffCSP** (Jiao et al., 2023) and **MatterGen** (Zeni et al., 2025), two leading diffusion-based approaches, and **FlowMM** (Miller et al., 2024), a high-performance flow-matching model. For fairness, we adopt the non-pretrained MatterGen and scale its parameter size to be comparable to the remaining models. All methods are originally tailored to perfectly ordered crystals, and cannot straightly process probabilistic compositions or multiple fractional coordinates. We therefore introduce systematic adaptations to extend their applicability to our disordered datasets.

For **SD**, we design three variants to handle probabilistic composition vectors: (i) *Sample*, which stochastically samples an element from the distribution defined by $s_i$; (ii) *Max*, which deterministically selects the element with the highest probability (*i.e.*, argmax); and (iii) *Prob*, which directly feeds the continuous probability vector into the model. We denote these variants by suffixes (e.g., DiffCSP-Max). With regard to **PD**, where a single site corresponds to two fractional coordinates, we modify the input graphs by representing each position as an independent node. This ensures that positional information is preserved, albeit at the cost of increasing the number of nodes and potentially complicating the structural representation. Further adaptation details are provided in Appendix D.2.

**Evaluation Metrics.** We adopt two standard metrics. The *Match Rate* (MR) measures the fraction of generated structures that can be matched to the ground truth using the `StructureMatcher` tool in `Pymatgen` (Ong et al., 2013), with tolerance parameters following those used in DiffCSP (Jiao et al., 2023). The *Root Mean Square Error* (RMSE) of atomic positions quantifies structural accuracy and is computed only for matched structures.

**Results.** The results in Table 1 reveal three key findings. **(1) Baseline hierarchy.** Across three baselines, the *Prob* variant consistently outperforms *Max* and *Sample*, as it retains the full probability vector rather than discarding information through stochastic sampling or deterministic hard assignments. **(2) DMFlow vs. baselines.** On pure SD datasets, DMFlow yields identical results to FlowMM-Prob by design. While on mixed-disorder SPD datasets, DMFlow achieves clear improvements. For instance, on COD-SPD-50, it obtains a Match Rate of 45.87% with an RMSE of 0.0725, compared to FlowMM-Prob's 44.00% and 0.0887. A similar advantage is observed on COD-SPD-20. These results underscore that DMFlow's unified treatment of disorder provides a critical advantage once positional disorder is introduced. It is also worth noting that although MatterGen achieves a lower RMSE, it exhibits a significantly inferior MR. This trade-off is expected due to the selection bias in RMSE calculation: a higher MR implies capturing more "borderline" samples with larger deviations, naturally inflating the average error compared to methods that only match the easiest subset. Conventionally, a higher MR is favored as it signifies a higher success rate in recovering ground-truth structures. Conversely, a low RMSE derived from a truncated matched set is of limited practical utility, as it indicates the model fails to align a significant portion of the dataset. **(3) SD vs. SPD difficulty.** All models degrade when moving from SD to SPD datasets, reflecting the challenge of modeling both disorder types. Yet DMFlow drops less, such as 3.25 vs.

Table 1: CSP performance on datasets containing up to 20 and 50 atoms, respectively. Note that on SD datasets, DMFlow and FlowMM-Prob achieve identical results, as positional disorder is absent and DMFlow reduces to the same formulation as FlowMM-Prob.

| Model | COD-SD-20 | | COD-SPD-20 | | COD-SD-50 | | COD-SPD-50 | |
|---|---|---|---|---|---|---|---|---|
| | MR (%) ↑ | RMSE ↓ | MR (%) ↑ | RMSE ↓ | MR (%) ↑ | RMSE ↓ | MR (%) ↑ | RMSE ↓ |
| DiffCSP-Sample | 49.73 | 0.0569 | 45.43 | 0.0800 | 32.63 | 0.0886 | 24.59 | 0.0895 |
| DiffCSP-Max | 59.54 | 0.0680 | 51.90 | 0.0681 | 40.10 | 0.0739 | 30.72 | 0.0750 |
| DiffCSP-Prob | 63.92 | 0.0675 | 53.09 | 0.0548 | 45.05 | 0.0509 | 35.48 | 0.0631 |
| MatterGen-Sample | 37.65 | 0.0585 | 32.73 | 0.0801 | 23.73 | 0.0643 | 19.23 | 0.0612 |
| MatterGen-Max | 50.26 | **0.0420** | 49.83 | 0.0487 | 30.21 | 0.0456 | 20.85 | **0.0488** |
| MatterGen-Prob | 53.87 | 0.0434 | 51.36 | **0.0273** | 37.14 | **0.0434** | 26.97 | 0.0517 |
| FlowMM-Sample | 55.16 | 0.0753 | 53.90 | 0.0865 | 38.13 | 0.0903 | 33.92 | 0.0811 |
| FlowMM-Max | 60.42 | 0.0799 | 58.14 | 0.0710 | 45.16 | 0.0897 | 40.81 | 0.0757 |
| FlowMM-Prob | 70.57 | 0.0738 | 64.16 | 0.0724 | 49.12 | 0.0681 | 44.00 | 0.0887 |
| DMFlow | **70.57** | 0.0738 | **65.30** | 0.0533 | **49.12** | 0.0681 | **45.87** | 0.0725 |

Table 2: DNG performance of DMFlow across four datasets. Rows highlighted in gray correspond to our proposed model, and N/A indicates that the module is not applied to the given dataset.

| Dataset | Configurable Modules | | Validity (%) ↑ | | Coverage (%) ↑ | | Property ↓ | |
|---|---|---|---|---|---|---|---|---|
| | Prob. Modeling | Multi-Interact. | Struc. | Comp. | Recall | Precision | Wdist ($\rho$) | Wdist ($N_{el}$) |
| COD-SD-20 | L1-Norm | N/A | 78.28 | 29.63 | 97.89 | 91.00 | 0.782 | 2.294 |
| | Softmax | N/A | 78.94 | 22.69 | 97.54 | 63.07 | 2.588 | 1.260 |
| | Simplex | N/A | **88.14** | **69.06** | **98.59** | **93.55** | **0.155** | **0.691** |
| COD-SPD-20 | L1-Norm | ✓ | 69.62 | 54.99 | 96.41 | 90.56 | 0.695 | 1.687 |
| | Softmax | ✓ | 69.57 | 56.34 | 96.09 | 89.54 | 1.236 | 1.990 |
| | Simplex | ✗ | 72.54 | 69.32 | 96.09 | 94.12 | 0.284 | **0.776** |
| | Simplex | ✓ | **87.30** | **69.99** | **99.18** | **94.31** | **0.259** | 0.825 |
| COD-SD-50 | L1-Norm | N/A | 81.23 | 36.41 | 94.72 | 94.15 | 0.257 | 3.301 |
| | Softmax | N/A | 81.57 | 20.98 | 97.25 | 68.62 | 2.618 | 2.354 |
| | Simplex | N/A | **90.84** | **68.57** | **97.30** | **94.75** | **0.151** | **1.583** |
| COD-SPD-50 | L1-Norm | ✓ | 53.73 | 52.72 | 94.63 | 92.56 | 1.232 | 1.762 |
| | Softmax | ✓ | 52.98 | 53.54 | 94.12 | 91.98 | 1.075 | 1.853 |
| | Simplex | ✗ | 76.93 | 64.48 | 97.19 | 93.54 | 0.240 | **1.343** |
| | Simplex | ✓ | **86.02** | **68.15** | **97.87** | **96.10** | **0.233** | 1.531 |

5.12 points for FlowMM-Prob on the 50-atom benchmark, underscoring the robustness of its unified framework over graph-splitting baselines.

## 5.2 De Novo Generation for Disordered Crystals

**Evaluation Metrics.** We adopt a suite of metrics to jointly assess the quality and diversity of generated samples. Quality is measured by *structural validity* (without clashes) and *compositional validity* (electrical neutrality). Diversity is evaluated using recall and precision, along with Wasserstein distances between generated and reference distributions of key material properties, crystal density ($\rho$) and the number of element types ($N_{el}$). Several of these metrics, however, are not directly applicable to disordered systems with fractional occupancies. To overcome this, we generalize the metrics to account for probabilistic site assignments, as detailed in Appendix D.3.

**Baselines.** Unlike CSP, where ordered baselines can be adapted, existing models are unable to perform DNG in the disordered setting. They can only generate ordered crystals with one-hot element assignments and thus cannot generate multiple compositions. Consequently, our comparison focuses on ablations of DMFlow itself. We investigate two critical components: (i) *Simplex constraints*. To validate its necessity, we replace it with unconstrained flow matching in Euclidean space. The generated raw weights are then normalized to sum to one using either L1 normalization (dividing by the vector sum) or Softmax normalization, followed by the same ensemble voting discretization (shown in Section 4.4); (ii) *Multiple interactions*, which capture both distance and direction embeddings

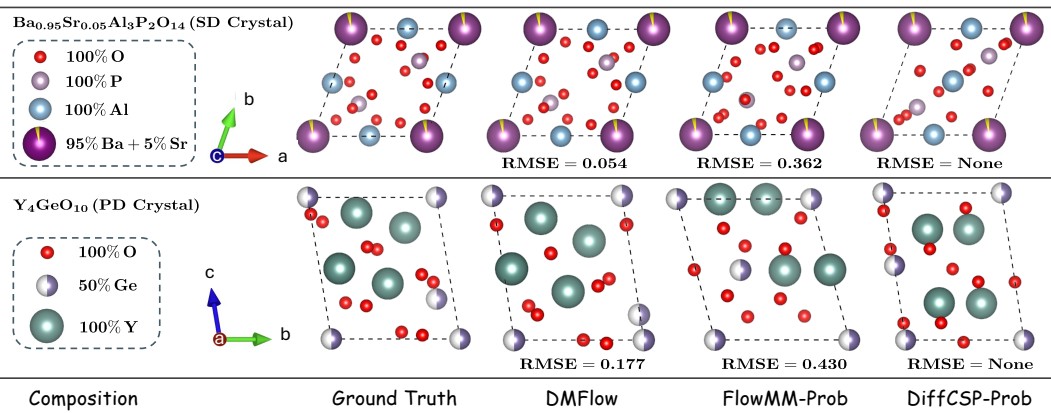

Figure 4: Visualization comparison of DMFlow and baseline models on the CSP task, covering both SD and PD cases. Here, RMSE = None indicates a failed structure prediction.

across all positional pairs (Eq. (13) and Eq. (14)). For ablation, we design a *single-interaction* variant: the two positional coordinates of each site are first averaged into a single coordinate using the positional weights $w_i$, and edge features are then computed as usual.

**Results.** Table 2 highlights three main observations. **First**, the simplex constraint is crucial for compositional realism. Replacing the Riemannian flow on the simplex with unconstrained Euclidean modeling, followed by L1 or Softmax normalization, results in a sharp decline in performance. Specifically, on COD-SD-20, compositional validity drops from 69.06% (Simplex) to 29.63% (L1-Norm) and 22.69% (Softmax). Furthermore, these unconstrained baselines yield distributions with significant property mismatch, indicated by consistently larger Wasserstein distances across all datasets. **Second**, the design of geometric interactions also matters. The multiple-interaction formulation explicitly models all positional pairs, resulting in clear improvements. On COD-SPD-50, structural validity rises from 76.93% to 86.02% and precision from 93.54% to 96.10%, while both property distances decrease. **Finally**, when both modules are enabled, DMFlow achieves the best performance across all benchmarks. It combines high validity and coverage with relatively lower Wasserstein distances. Taken together, these findings confirm that simplex-constrained flow matching and multiple-interaction modeling play complementary roles, and that their integration is essential for accurate and robust disordered crystal generation.

## 5.3 MORE ANALYSES

**Visualization of Generated Structures.** We present qualitative visualizations of generated crystal structures to illustrate how different models capture SD and PD in practice. Specifically, we compare DM-Flow, FlowMM-Prob, and DiffCSP-Prob on the COD-SPD-20 dataset, using representative cases for both SD and PD scenarios (see Fig. 4). In the SD case, DM-

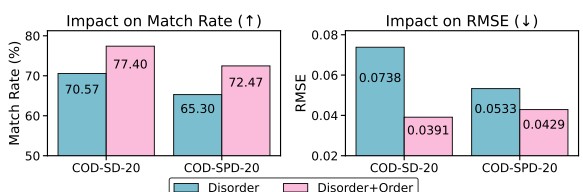

Figure 5: CSP performance with data augmentation.

Flow almost perfectly reconstructs the ground-truth structure, achieving a very low RMSE of 0.054, whereas the baseline models show some mismatches in atomic positions. In the more challenging PD case, DMFlow's RMSE increases slightly, yet the major atomic species (e.g., Y and Ge) are generated with reasonable accuracy. By contrast, baselines exhibit severe mismatches, failing to reproduce key local arrangements. These comparisons highlight the efficacy of DMFlow and demonstrate the advantage of our unified formulation over existing baselines. In Appendix D.4, we further visualize the flow-matching trajectory to provide insights into how DMFlow progressively generates disordered crystals.

**Augmentation with Ordered Crystals.** A significant challenge in modeling disordered materials is the relative scarcity of data compared to the vast databases of ordered crystals. Thanks to our

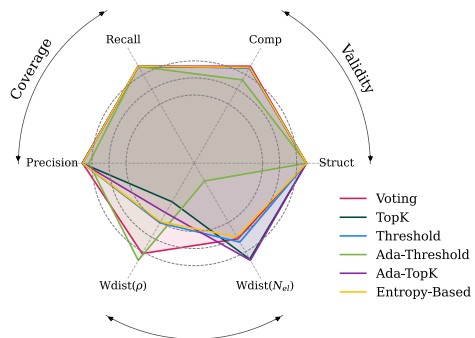 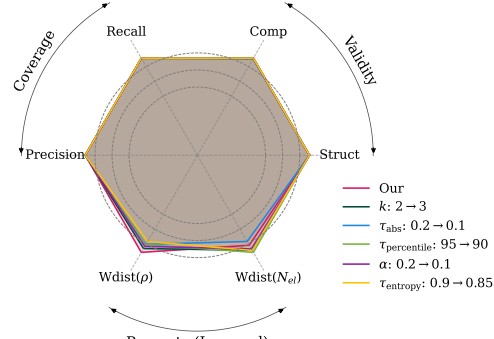

Figure 6: Analysis of the discretization strategy on COD-SD-20. (a) Comparison between the proposed ensemble voting and individual heuristics. (b) Sensitivity analysis under hyperparameter perturbations. *Note:* Since property metrics (Wdist) are lower-is-better while others are higher-is-better, we visualize their reciprocal values to ensure that a larger span consistently represents better performance across all axes.

unified representation, we can augment "COD-*-20" with ordered samples from MP-20 (Jain et al., 2013) for training and validation (while keeping test sets fixed). For the CSP task, adding ordered data yields clear benefits. As shown in Fig. 5, on COD-SD-20 the Match Rate rises from 70.57% to 77.40% while RMSE drops from 0.0738 to 0.0391, with a similar trend on COD-SPD-20. This setup evaluates ordered data augmentation, indicating that DMFlow unifies ordered and disordered structures and leverages ordered samples to improve reconstruction quality on disordered test sets. Additional performance gains on the DNG task are reported in Appendix D.5.

**Ablation on Discretization.** To validate the effectiveness of our discretization strategy, we compare the ensemble voting scheme against individual heuristics and analyze its sensitivity to hyperparameter settings. As shown in Fig. 6(a), individual heuristics exhibit distinct trade-offs; for instance, TopK selection suffers from poor density alignment (Wdist ($\rho$)) while Adaptive-Threshold and Entropy-Based selection yields inferior element count distributions (Wdist ($N_{el}$)). In contrast, our ensemble approach mitigates these specific biases, achieving the highest compositional validity and delivering consistently balanced performance across all metrics. Meanwhile, the sensitivity analysis in Fig. 6(b) reveals that the scheme is highly robust to hyperparameter perturbations. Specifically, while varying thresholds induces slight fluctuations in property statistics (bottom axes), the structural validity and coverage metrics (top axes) remain essentially unchanged. This confirms that the voting mechanism is stable and insensitive to moderate parameter tuning.

## 6 CONCLUSION

In this work, we present **DMFlow**, the first generative framework for disordered crystal structures, addressing the limitations of prior methods confined to ordered cases. At its core is a unified formulation that jointly encodes substitutional disorder, positional disorder, and ordered structures, enabling consistent generation across crystal types. On top of this, we design a flow-matching framework that generates structures and disorder weights on Riemannian manifolds with periodicity and simplex constraints, aided by a graph neural network that processes continuous disorder input and models multiple positional interactions. We also propose a voting strategy to discretize disorder weights into valid atomic assignments. Experiments on our constructed benchmark of disordered crystals show that DMFlow delivers state-of-the-art performance on both crystal structure prediction and *de novo* generation, validating the effectiveness of our unified formulation and model design.

## ETHICS STATEMENT

The authors have read and complied with the ICLR Code of Ethics. Our work aims to advance the state of the art in machine learning by introducing a novel methodology. The study does not involve human subjects or sensitive personal data. We recognize our responsibility to consider the broader

societal impacts of this research. While the intended applications are constructive, we acknowledge that powerful technologies may be susceptible to misuse or unintended consequences. To mitigate such risks, we have prioritized transparency and reproducibility in our experiments. We condemn any malicious use of our work and welcome feedback from the community on ethical aspects that may warrant further consideration.

## REPRODUCIBILITY STATEMENT

We are committed to ensuring the reproducibility of our research. To this end, we provide comprehensive details of our methodology, dataset construction, and experimental setup. A complete description of data preprocessing, evaluation metrics, and baselines can be found in Section 5. Additional implementation details, including all model hyperparameters and training configurations, are provided in Appendices C and D. Furthermore, we will release the source code publicly upon publication to facilitate replication and further research.

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

## A  THE USE OF LARGE LANGUAGE MODELS

Large Language Models (LLMs) were used solely for post-editing to enhance the language, clarity, and readability of this paper. The authors are responsible for all research ideas, content, and claims, as the LLM played no role in the scientific conception, data analysis, or interpretation. All text was thoroughly reviewed and approved by the authors.

## B  PRELIMINARIES OF FLOW MATCHING

### B.1  GENERAL FLOW MATHCING

**Conditional Flow Matching.** Flow matching aims to learn a time-dependent velocity field $v_t(x)$ that transports samples from a simple prior distribution $p_0(x)$ (e.g., a standard Gaussian) to a complex target data distribution $p_1(x)$. Ideally, one would train a neural network $v_\theta(x_t, t)$ to match the true vector field $u_t(x_t)$ of the marginal probability path $p_t(x_t)$ connecting $p_0$ and $p_1$. However, this marginal vector field $u_t(x_t)$ is generally unknown and intractable to compute. Conditional Flow Matching (CFM) (Lipman et al., 2023) elegantly circumvents this issue. Instead of modeling the complex marginal path, CFM defines a simpler conditional probability path $p_t(x_t|x_1)$ that connects a prior sample $x_0 \sim p_0$ to a single data sample $x_1 \sim p_1$. The key insight is that the marginal vector field is the expectation of the conditional vector field, $u_t(x) = \mathbb{E}_{p_1(x_1)}[u_t(x|x_1)]$. Therefore, one can train the model by minimizing the tractable objective, which regresses the learned velocity field onto the easily computed conditional vector field $u_t(x_t|x_1)$.

**Riemannian Flow Matching.** Generating physical systems like molecules or crystals requires respecting their inherent symmetries. Crystal structures are not simple vectors in Euclidean space; their properties often reside on non-Euclidean manifold $\mathcal{M}$. To properly handle these geometric constraints, we must formulate our model using the language of Riemannian geometry. At any point $x \in \mathcal{M}$, the manifold can be locally approximated by a vector space known as the tangent space, denoted $\mathcal{T}_x\mathcal{M}$. Our learned velocity field $v_t(x, t)$ is defined as a vector within this tangent space. The concept of a flow path on a manifold is generalized by the geodesic, which is the locally shortest path between two points. The connection between the manifold and its tangent spaces is established by two key operations: the exponential map $\exp_x : \mathcal{T}_x\mathcal{M} \to \mathcal{M}$ and its inverse, the logarithm map $\log_x : \mathcal{M} \to \mathcal{T}_x\mathcal{M}$. The exponential map takes a tangent vector $v \in \mathcal{T}_x\mathcal{M}$ and maps it to a point on the manifold by traveling along the geodesic starting at $x$ in the direction of $v$. Conversely, the logarithm map calculates the unique tangent vector at $x$ that corresponds to the geodesic path to another point $y$. With these tools, we can define the geodesic path between $x_0$ and $x_1$ by $x_t := \exp_{x_0}(t \cdot \log_{x_0}(x_1))$.

### B.2  FLOWMM FORMULATION

In this section, we provide the full details of the flow matching setup for the lattice parameters $\boldsymbol{L}$ and fractional coordinates $\boldsymbol{F}$ (including $\boldsymbol{F}'$ for positional disorder). These follow the design in FlowMM (Miller et al., 2024), but we include them here for completeness.

**Flow Matching on Lattice Parameters.** We parameterize the lattice matrix $\boldsymbol{L}$ via a rotation-invariant form, which is the Niggli-reduced lattice parameters $\boldsymbol{l} = (a, b, c, \alpha, \beta, \gamma)$, where $(a, b, c) \in \mathbb{R}^3_+, (\alpha, \beta, \gamma) \in [60, 120]^3$ denote the lengths and angles of the parallelepiped. To initialize from plausible cells, the prior distribution is factorized as $p_0(\boldsymbol{l}) = p_0(a, b, c)p_0(\alpha, \beta, \gamma)$ with $p_0(a, b, c) := \prod_{\eta \in \{\alpha, \beta, \gamma\}} \text{LogNormal}(\eta; \text{loc}_\eta, \text{scale}_\eta)$ and $p_0(\alpha, \beta, \gamma) := \mathcal{U}(60, 120)$. To avoid issues arised from the angular boundaries, each angle is further mapped into an unconstrained Euclidean space via $\varphi(\eta) = \text{logit}\left(\frac{\eta-60}{120}\right), \varphi^{-1}(\eta') = 120 \cdot \sigma(\eta') + 60$, where $\sigma(\cdot)$ is the sigmoid function. The resulting unconstrained representation is given by $\tilde{\boldsymbol{l}} = (a, b, c, \varphi(\alpha), \varphi(\beta), \varphi(\gamma))$. Given the interpolated samples $\tilde{\boldsymbol{l}}_t = (1-t)\tilde{\boldsymbol{l}}_0 + t\tilde{\boldsymbol{l}}_1$, the training objective on lattice is defined as:

$$\mathcal{L}_{\tilde{\boldsymbol{l}}} = \mathbb{E}_{t \in \mathcal{U}(0,1), p(\tilde{\boldsymbol{l}}_1), p(\tilde{\boldsymbol{l}}_t|\tilde{\boldsymbol{l}}_1)} \left[ \frac{1}{6} \|v_{\theta, \tilde{\boldsymbol{l}}}(\tilde{\boldsymbol{l}}_t, t) - (\tilde{\boldsymbol{l}}_1 - \tilde{\boldsymbol{l}}_0)\|_2^2 \right]. \tag{20}$$

**Flow Matching on Fractional Coordinates.** Let $\boldsymbol{F}_0 \sim \mathcal{U}(0,1)$ denote the sample from the uniform prior, the displacement from $\boldsymbol{F}_0$ to data $\boldsymbol{F}_1$ is obtained by the wrapped logarithmic map:

$$\boldsymbol{s}(\boldsymbol{F}_0, \boldsymbol{F}_1) = \log_{\boldsymbol{F}_0}(\boldsymbol{F}_1) = \mathrm{wrap}(\boldsymbol{F}_1 - \boldsymbol{F}_0), \tag{21}$$

where $\mathrm{wrap}(z) = (z + 0.5 \bmod 1) - 0.5$ applies the shortest periodic shift to each component. The corresponding exponential map reconstructs $\boldsymbol{F}_1$ from $\boldsymbol{F}_0$ and its displacement vector as:

$$\exp_{\boldsymbol{F}_0}(\boldsymbol{s}(\boldsymbol{F}_0, \boldsymbol{F}_1)) = (\boldsymbol{F}_0 + \boldsymbol{s}(\boldsymbol{F}_0, \boldsymbol{F}_1)) \bmod 1. \tag{22}$$

To remove the effect of the global translation, we subtract the mean displacement among all atoms and correct the logarithmic map as: $\hat{\boldsymbol{s}}(\boldsymbol{F}_0, \boldsymbol{F}_1) = \boldsymbol{s}(\boldsymbol{F}_0, \boldsymbol{F}_1) - \frac{1}{N}\sum_{i=1}^{N} \boldsymbol{s}_i(\boldsymbol{F}_0, \boldsymbol{F}_1)$. The geodesic interpolation between $\boldsymbol{F}_0$ and $\boldsymbol{F}_1$ is then given by: $\boldsymbol{F}_t = \exp_{\boldsymbol{F}_0}(t \cdot \hat{\boldsymbol{s}}(\boldsymbol{F}_0, \boldsymbol{F}_1)), t \in [0,1]$. The training objective minimizes the error between the predicted velocity field and the normalized displacement:

$$\mathcal{L}_{\boldsymbol{F}} = \mathbb{E}_{t \in \mathcal{U}(0,1), p(\boldsymbol{F}_1), p(\boldsymbol{F}_t|\boldsymbol{F}_1)} \left[ \frac{1}{3N} \big\| v_{\theta, \boldsymbol{F}}(\boldsymbol{F}_t, \boldsymbol{A}) - \hat{\boldsymbol{s}}(\boldsymbol{F}_0, \boldsymbol{F}_1) \big\|_2^2 \right]. \tag{23}$$

For sites with PD, the secondary coordinates $\boldsymbol{F}'$ are modeled identically as $\boldsymbol{F}$ with the same wrapped geodesics and training objective.

## C   DETAILS ON THE DISCRETIZATION ALGORITHM

This section provides a detailed description of the two-stage discretization algorithm used to convert continuous substitutional disorder vectors $\boldsymbol{s}_i$ into multi-hot vectors representing discrete atomic compositions.

**Stage I: Ordered Site Identification.**   For a given disorder vector $\boldsymbol{s}_i$, let

$$\begin{aligned} j_1 &= \arg\max_j s_{i,j}, \quad p_1 = s_{i,j_1}, \\ j_2 &= \arg\max_{j \neq j_1} s_{i,j}, \quad p_2 = s_{i,j_2}. \end{aligned} \tag{24}$$

The site $i$ is classified as ordered if $\frac{p_1}{p_2} > \tau_{\mathrm{ratio}}$, where $\tau_{\mathrm{ratio}}$ is set to 3.0 in our experiments. If the condition is satisfied, the final vector is given by the one-hot representation corresponding to index $j_1$.

**Stage II: Ensemble Voting for Disordered Sites.**   If a site is classified as disordered, we generate five candidate multi-hot vectors, $\{\boldsymbol{v}_1, \ldots, \boldsymbol{v}_5\}$, using the following heuristic methods:

1. **Top-k Selection:** The multi-hot vector $\boldsymbol{v}_1$ is formed by selecting the $k$ elements with the highest probabilities in $\boldsymbol{s}_i$. Here $k$ is a fixed hyperparameter and we set it to 2.

2. **Absolute Thresholding:** An element $j$ is selected if its probability $s_{i,j}$ exceeds a fixed threshold:
$$\boldsymbol{v}_{2,j} = \mathbb{I}(s_{i,j} > \tau_{\mathrm{abs}}), \quad \tau_{\mathrm{abs}} = 0.2. \tag{25}$$

3. **Percentile Thresholding:** An element $j$ is selected if $s_{i,j}$ lies within the top $\tau_{\mathrm{percentile}}$ percentile of $\boldsymbol{s}_i$:
$$\tau_{\mathrm{percentile}} = 95. \tag{26}$$

4. **Adaptive Thresholding:** Select elements whose probabilities exceed a fraction of the maximum probability:
$$\boldsymbol{v}_{4,j} = \mathbb{I}(s_{i,j} > \alpha \cdot p_1), \quad \alpha = 0.2. \tag{27}$$

5. **Entropy-Based Selection:** The selection is based on the normalized Shannon entropy (Lin, 2002):
$$H(\boldsymbol{s}_i) = -\sum_j s_{i,j} \log s_{i,j}, \quad \hat{H} = \frac{H(\boldsymbol{s}_i)}{\log d}, \tag{28}$$

   where $d$ is the number of atomic categories. If $\hat{H} > \tau_{\mathrm{entropy}}$ ($\tau_{\mathrm{entropy}} = 0.9$), we conservatively select only the argmax element. Otherwise, we apply the adaptive rule with $\alpha = 0.2$.

**Final Voting.** The five candidate vectors are aggregated into a vote count vector $\boldsymbol{v}^{(c)} = \sum_{m=1}^{5} \boldsymbol{v}_m$. The final discrete multi-hot vector for site $i$, denoted $\boldsymbol{z}_i$, is then determined by applying a minimum vote threshold:

$$z_{i,j} = \mathbb{I}(v_j^{(c)} \geq \tau_{\text{vote}}), \tag{29}$$

where $z_{i,j}$ is the $j$-th component of the vector $\boldsymbol{z}_i$, and $\tau_{\text{vote}}$ is a hyperparameter set to 4.

## D    EXPERIMENTAL DETAILS

This section first describes the training setup and hyperparameters of DMFlow, followed by the implementation of baseline methods. We then introduce generalized evaluation metrics for disordered crystals and visualize flow-matching trajectories. Finally, we present ablation studies on loss weighting and examine the effect of ordered data augmentation on the DNG task.

### D.1    TRAINING AND MODEL HYPERPARAMETERS

The specific hyperparameter settings used in our experiments are summarized in Table 3. The terms $\tilde{\lambda}_\square$ indicate the relative weights assigned to different components of the loss function, which are normalized during training and denoted by $\lambda_\square$.

Table 3: Hyperparameter settings for model architecture, training, and sampling.

| Hyperparameter | Value |
| --- | --- |
| Hidden Dimensions (`hidden_dim`) | 512 |
| Number of GNN Layers (`num_layers`) | 6 |
| Activation Function | SiLU |
| Optimizer | Adam |
| Learning Rate | $6 \times 10^{-4}$ |
| Epochs | 2000 |
| Batch Size | 512 (300 for COD-SPD-50) |
| Loss Weights (CSP Task) | $\tilde{\lambda}_{\boldsymbol{F}}$: 400, $\tilde{\lambda}_{\tilde{\boldsymbol{i}}}$: 1, $\tilde{\lambda}_{\boldsymbol{F}'}$: 40 |
| Loss Weights (DNG Task) | $\tilde{\lambda}_{\boldsymbol{S}}$: 2000, $\tilde{\lambda}_{m\boldsymbol{F}}$: 400, $\tilde{\lambda}_{\tilde{\boldsymbol{i}}}$: 1, $\tilde{\lambda}_{\boldsymbol{W}}$: 40, $\tilde{\lambda}_{\boldsymbol{F}'}$: 40 |
| ODE Integration Steps | 1000 |
| Anti-Annealing Slope | 20 |

### D.2    IMPLEMENTATION DETAILS FOR BASELINES

This section provides additional details on the specific modifications made to the baseline models, DiffCSP (Jiao et al., 2023), MatterGen (Zeni et al., 2025) and FlowMM (Miller et al., 2024), to adapt them for our disordered datasets. The adaptations can be categorized into two main areas: how the input data is processed and the modifications to the model architecture itself.

#### D.2.1    ADAPTATIONS FOR INPUT DATA PROCESSING

A primary challenge was to create a unified input pipeline that could handle the various feature types required by our baseline variants.

**Unified Input Layer.** The original implementations of DiffCSP and MatterGen utilize an `nn.Embedding` layer, which is highly efficient for one-hot encoded element types. However, to accommodate our three adaptation variants (*Sample*, *Max*, and *Prob*) in a single framework, we replaced this layer in both baselines with a standard `nn.Linear` layer. An `nn.Linear` layer is more general: its behavior is equivalent to a lookup operation for one-hot inputs (*Sample*, *Max*) but, crucially, it can also naturally process the continuous vectors from the *Prob* variant.

**Feature Construction for Positional Disorder.** When adapting for PD structures, a site $i$ with two positions $(\boldsymbol{f}_i, \boldsymbol{f}'_i)$ and occupancies $\boldsymbol{w}_i = [w_{i,0}, w_{i,1}]^\top$ is split into two separate nodes. To preserve

the occupancy information, we scale the site's original D-dimensional substitutional disorder vector, $s_i$, to create the feature vectors for the new nodes:

- The first node (at position $f_i$) receives the feature vector $\hat{s}_i = w_{i,0} \cdot s_i$.
- The second node (at position $f_i'$) receives the feature vector $\hat{s}_i' = w_{i,1} \cdot s_i$.

It is important to note that these resulting feature vectors, $\hat{s}_i$ and $\hat{s}_i'$, are no longer probability distributions residing on a simplex; their components sum to $w_{i,0}$ and $w_{i,1}$, respectively. This heuristic is necessary to encode the site weight information for the baseline models, which lack a native mechanism to handle positional disorder. These scaled vectors are then processed by the unified `nn.Linear` input layer described previously.

### D.2.2 ARCHITECTURAL MODIFICATIONS

The data adaptation for PD had a direct consequence on the model architecture that required a specific modification. Our strategy of splitting PD sites into independent nodes increases the total number of nodes (atoms) in the input graph. The official source code for FlowMM hardcodes a maximum limit of 100 atoms for its atom-count embedding. We found that several structures in our dataset exceeded this limit after splitting, which would cause a runtime error. To ensure all structures could be evaluated fairly, we modified the its architecture by increasing this upper limit to 150.

For MatterGen, we configure the `hidden_dim` $= 384$ and `num_layers` $= 2$ so that its parameter size remains comparable to the other models, yielding a 14.2M-parameter model. Moreover, we train MatterGen from scratch instead of using its pretrained checkpoints for fairness.

### D.3 GENERALIZED EVALUATION METRICS FOR DISORDERED STRUCTURES

As mentioned in Section 5.2, several standard metrics used in the evaluation of generative models for ordered crystals are not directly applicable to disordered structures with fractional occupancies. This section details the modifications we implemented to create a robust and fair evaluation protocol for the De Novo Generation (DNG) task.

**Compositional Validity.** The compositional validity metric typically checks if a generated chemical formula is charge-neutral. Standard implementations of this check are designed to operate on formulas with integer atom counts (e.g., $Fe_2O_3$). To handle our generated structures, which have probabilistic compositions, we extended the validation algorithm to support fractional atomic counts. This allows it to correctly assess the charge neutrality of expected chemical formulas derived from disordered materials.

**Wasserstein Distance of Element Types Distribution.** This metric (wdist ($N_{el}$)) evaluates the Wasserstein distance between the distributions of the count of distinct element types per structure in the generated and reference sets. In an ordered crystal, this count is a simple integer obtained by identifying the unique elements. However, a single site in a disordered structure can be fractionally occupied by multiple different elements. Our adaptation involves a straightforward modification to the counting logic to correctly identify the complete set of unique element species present in a structure, accounting for all elements listed across all probabilistic site occupancies. As a result of this phenomenon, the values for $N_{el}$ are generally higher for disordered materials compared to their ordered counterparts, as a single site can contribute multiple elements to the total count.

**Coverage Metrics (Recall and Precision).** The coverage metrics rely on structural fingerprints to compare the similarity between generated and reference crystal structures. However, the underlying fingerprinting libraries (e.g., from Matminer (Ward et al., 2018)) can only process ordered structures and raise an error when encountering a disordered site with multiple, fractionally-occupied elements.

To overcome this limitation, we developed a stochastic fingerprinting protocol to compute a representative fingerprint for any given disordered structure. The procedure is as follows:

1. For a single disordered structure, we generate a discrete, ordered realization by sampling a specific element for each disordered site according to the probabilities in its occupancy vector.

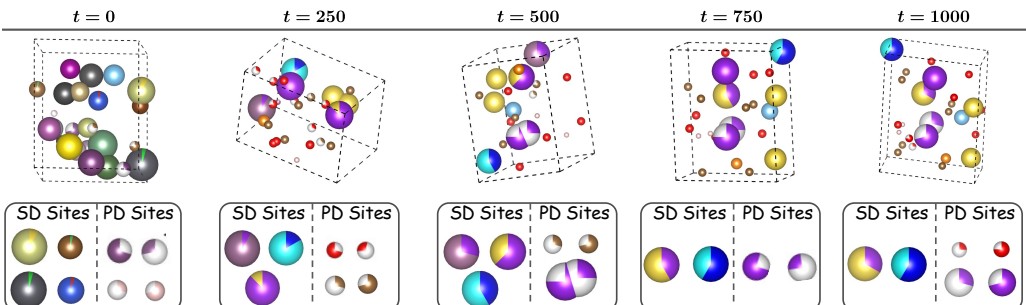

Figure 7: Visualization of the flow-matching trajectory at different steps.

2. We repeat this stochastic sampling process 10 times, creating an ensemble of 10 distinct ordered structures, each representing a plausible snapshot of the disordered material.

3. The structural fingerprint is computed for each of these 10 ordered realizations.

4. Finally, we average the 10 resulting fingerprint vectors to produce a single, robust fingerprint that represents the expected structural features of the original disordered crystal.

This averaged fingerprint is then used in the standard way to calculate the recall and precision metrics, enabling a fair comparison of structural diversity and coverage for disordered systems.

### D.4 FLOW-MATCHING TRAJECTORY VISUALIZATION

As shown in Fig. 7, we visualize representative structures from COD-SPD at flow-matching steps 0, 250, 500, 750, and 1000 to illustrate the full generation dynamics. At the beginning ($t = 0$), the system is highly chaotic: atoms of many different element types are placed without clear order, and severe overlaps between atoms are observed due to the lack of spatial organization. After a few hundred steps ($t = 250$ and $t = 500$), the structure begins to organize. Substitutional disorder starts to settle into meaningful probabilistic patterns, and the atomic coordinates gradually spread out, alleviating the initial collisions. By $t = 750$, the generated structure already shows clear crystallographic order, with substitutional disorder reaching a stable state and positional disorder becoming distinguishable. At the final step ($t = 1000$), DMFlow produces a coherent crystal that faithfully combines both substitutional (SD) and positional disorder (PD): SD sites converge to realistic element distributions, while PD sites manifest as distinct alternative positions.

This progressive trajectory demonstrates how DMFlow resolves both compositional and positional uncertainty in a smooth and interpretable manner. Unlike baseline models unable to generate meaningfully disordered crystals from scratch, our method gradually transforms a random initialization into a valid disordered crystal. Such visualizations not only highlight the stability of the flow-matching process but also provide intuitive evidence that DMFlow can simultaneously capture the complexity of substitutional and positional disorder during generation.

### D.5 ORDERED DATA AUGMENTATION ON DNG

To evaluate whether ordered data can enhance the DNG performance of DMFlow, we compare the model trained with and without ordered augmentation on COD-SD-20 and COD-SPD-20. As shown in Fig. 8, the augmented model shows clear improvements in several aspects: for example, structural validity rises from 88.14% to 98.60% on COD-SD-20, and compositional validity increases similarly across both datasets. Coverage recall remains nearly perfect, while precision exhibits a moderate decrease, reflecting a broader exploration of candidate structures. For property alignment, the Wasserstein distance on element counts is notably reduced, whereas the distance on densities ($\rho$) increases, which may be attributed to distributional differences between the added ordered structures and the original disordered datasets. Overall, incorporating ordered structures enables DMFlow to generate more valid and compositionally consistent disordered crystals, at the cost of a modest shift in density distribution.

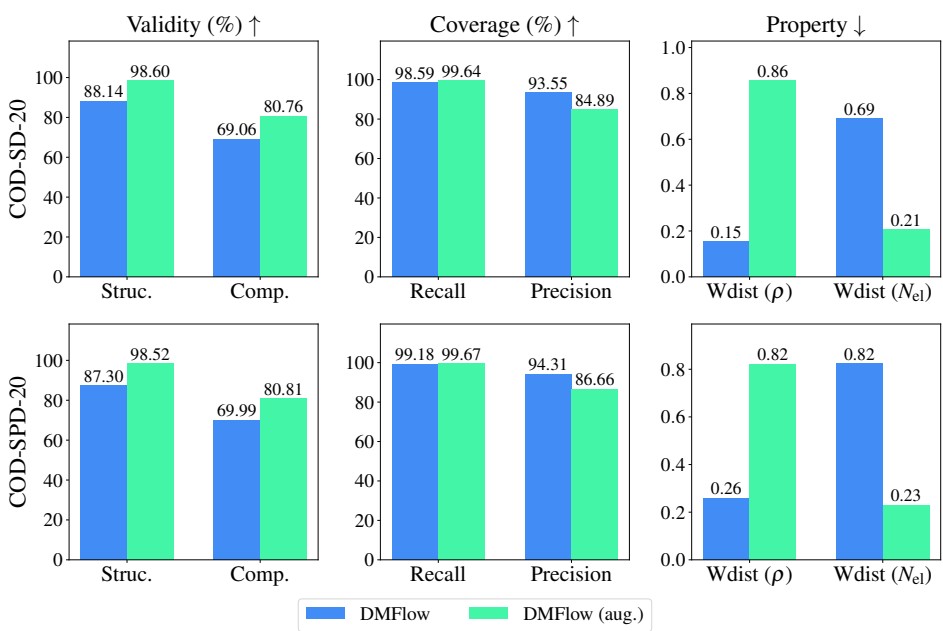

Figure 8: DNG performance with ordered crystals augmentation.

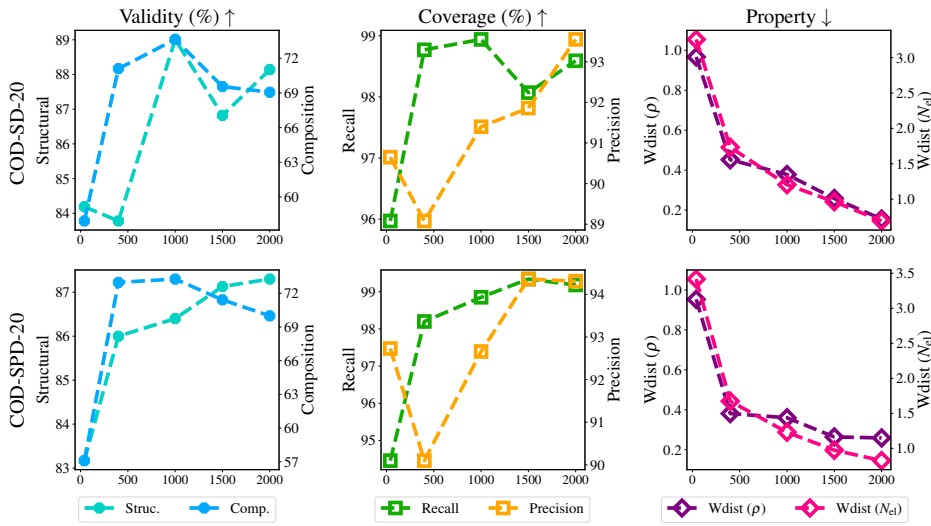

Figure 9: Ablation on the substitutional loss weight in the DNG task.

## D.6 ABLATION ON LOSS WEIGHTING

In the DNG task, we perform an ablation study on the loss weighting of the substitutional vector, denoted as `cost_type`. This term controls the relative weight assigned to the loss for generating substitutional disorder, and we examine its effect on COD-SD-20 and COD-SPD-20. Specifically, we vary `cost_type` across $\{40, 400, 1000, 1500, 2000\}$ and report the resulting metrics in Fig. 9. It is observed that increasing the weight generally leads to improvements across all metrics, despite minor fluctuations. The overall trend suggests that stronger supervision on the substitutional vector facilitates more accurate generation. Accordingly, we adopt 2000 as the default setting in all reported experiments.

# E EXTENSION TO HIGHER-ORDER POSITIONAL DISORDER

In the main text, we focused on binary Positional Disorder (PD), where a site splits into at most two positions. This choice was motivated by the prevalence of binary cases in the COD dataset (Gražulis et al., 2009). However, the unified framework of DMFlow is naturally extensible to higher-order PD, where a single crystallographic site can be probabilistically occupied by more than two split positions. In this section, we detail the generalization of our representation, network architecture, and dataset construction to handle such cases, and provide empirical evaluations.

## E.1 GENERALIZED REPRESENTATION AND FLOW MATCHING

To accommodate PD of arbitrary complexity, we define a maximum PD order, denoted as $\ell_{\max}$. The representation of a site $i$ is generalized from the binary tuple to a multi-component set. The positional weights are now represented by a vector on a higher-dimensional simplex, $\hat{\boldsymbol{w}}_i \in \Delta^{\ell_{\max}-1}$, such that $\sum_{\ell=0}^{\ell_{\max}-1} \hat{w}_{i,\ell} = 1$. Correspondingly, the fractional coordinates are extended to a multi-channel matrix $\hat{\boldsymbol{F}}_i \in [0,1)^{\ell_{\max} \times 3}$, where the $\ell$-th row corresponds to the coordinate of the $\ell$-th split position. For sites with an actual PD order less than $\ell_{\max}$, we pad the unused weight entries and the coordinate entries with zeros (which do not affect the physics due to zero weights).

When performing flow matching, we flatten the coordinate matrix $\hat{\boldsymbol{F}}_i$ into a vector of dimension $3\ell_{\max}$. Theoretically, the manifold of $\ell_{\max}$ split positions can be viewed as a product manifold $\mathcal{M}^{\ell_{\max}} = \mathbb{T}^3 \times \cdots \times \mathbb{T}^3$ ($\ell_{\max}$ times). For product manifolds equipped with a product metric, the geodesic distance decomposes into the sum of squared distances on the factor manifolds (Lee, 2018). Therefore, performing CFM on the flattened vector in $[0,1)^{3\ell_{\max}}$ is mathematically equivalent to defining a joint flow on the product manifold. This allows us to utilize the same training objective as defined in Eq. (23), simply scaling the dimensions of the input and output heads.

## E.2 ARCHITECTURE MODIFICATION: WEIGHTED INTERACTION SUMMATION

The velocity prediction network described in Section 4.3 relies on edge embeddings that explicitly enumerate PD state combinations. In the binary case, we concatenated embeddings for all $2 \times 2 = 4$ pairs (Eqs. (13) and (14)). For higher-order PD, the number of pairs becomes $\ell_{\max}^2$, which renders concatenation computationally prohibitive and effectively sparse (as many weights are zero).

To address this, we modify the edge embedding aggregation strategy from concatenation to *weighted summation*. We first compute the geometric features for all $\ell_{\max} \times \ell_{\max}$ pairwise combinations between site $i$ and site $j$, and then aggregate them weighted by their joint occupancy probabilities. The generalized formulas for distance and direction embeddings are:

$$\hat{e}_{\text{dist}}^{ij} = \sum_{\ell_1=0}^{\ell_{\max}-1} \sum_{\ell_2=0}^{\ell_{\max}-1} \left( \hat{w}_{i,\ell_1} \hat{w}_{j,\ell_2} \cdot \text{SinusoidalEmb}\left( \log_{\hat{\boldsymbol{F}}_i[\ell_1,:]}(\hat{\boldsymbol{F}}_j[\ell_2,:]) \right) \right), \quad (30)$$

$$\hat{e}_{\text{dir}}^{ij} = \sum_{\ell_1=0}^{\ell_{\max}-1} \sum_{\ell_2=0}^{\ell_{\max}-1} \left( \hat{w}_{i,\ell_1} \hat{w}_{j,\ell_2} \cdot \frac{M(l)(\hat{\boldsymbol{F}}_j[\ell_2,:] - \hat{\boldsymbol{F}}_i[\ell_1,:])}{\|M(l)(\hat{\boldsymbol{F}}_j[\ell_2,:] - \hat{\boldsymbol{F}}_i[\ell_1,:])\|} \right). \quad (31)$$

Here, $\hat{\boldsymbol{F}}_i[\ell_1,:]$ takes the $\ell_1$-row of the $\hat{\boldsymbol{F}}_i$ matrix. This formulation is permutation invariant with respect to the ordering of split positions and naturally handles variable PD orders; for non-existent positions, $\hat{w}_{\cdot,\cdot} = 0$, ensuring they contribute zero to the message passing.

## E.3 DATASET CONSTRUCTION: COD-SHPD

To evaluate the model on higher-order disorder, we constructed a new dataset, termed **COD-SHPD** (Substitutional and Higher-order Positional Disorder). We filtered the COD database for structures exhibiting PD with up to $\ell_{\max} = 5$ split positions. To increase data diversity, we also relaxed the maximum atom count constraint from 50 to 80 atoms per unit cell. These higher-order PD structures were combined with the existing SD structures from COD-SD.

Table 4: DGN performance on the COD-SHPD dataset ($\ell_{\max} = 5$). "Without simplex" denotes using L1 Normalization.

| Dataset | Configurable Modules | | Validity (%) ↑ | | Coverage (%) ↑ | | Property ↓ | |
|---|---|---|---|---|---|---|---|---|
| | Simplex | Multi-Interact. | Struc. | Comp. | Recall | Precision | Wdist ($\rho$) | Wdist ($N_{el}$) |
| COD-SHPD | ✗ | ✓ | 25.61 | 32.99 | 68.85 | 76.44 | 1.939 | 0.898 |
| | ✓ | ✗ | 22.80 | 63.10 | 67.96 | 95.95 | **0.452** | **0.713** |
| | ✓ | ✓ | **48.16** | **63.46** | **97.13** | **97.10** | 0.731 | 1.146 |

The resulting dataset contains a total of 14,593 structures (5,497 with higher-order PD). We applied a random split of 80%/10%/10%, resulting in 11,668 training samples, 1,461 validation samples, and 1,464 test samples.

### E.4 RESULTS AND ANALYSIS

We trained DMFlow with the generalized architecture on COD-SHPD and evaluated its performance on the De Novo Generation (DNG) task. The results are summarized in Table 4.

**Performance Analysis.** As expected, extending the generation task to higher-order PD ($\ell_{\max} = 5$) and larger systems (80 atoms) introduces significant complexity, leading to a decrease in validity metrics compared to the binary cases reported in the main text. Nevertheless, DMFlow maintains a respectable performance, achieving 63.46% compositional validity and 48.16% structural validity. Notably, the model exhibits excellent coverage ($> 97\%$), indicating it effectively captures the diversity of this complex data distribution. Property alignment remains reasonable, though the higher Wasserstein distances reflect the increased difficulty in matching the statistics of complex disordered structures.

**Ablation Studies.** We revisited the ablation studies in this generalized setting:

- **Impact of Simplex Constraint:** Consistent with our main results, removing the simplex constraint leads to a drastic drop in compositional validity (from 63.46% to 32.99%) and structural validity. This confirms that the Riemannian flow matching on the simplex is crucial for learning valid disorder weights, regardless of the PD order.

- **Impact of Multi-Interaction:** Removing the explicit modeling of all position pairs hampers the structural validity (dropping to 22.80%). This validates our generalized weighted summation design (Eqs. (30) and (31)), proving that explicit modeling of interactions between all split positions is essential for generating geometrically valid crystals. Notably, this configuration improves property statistics, likely because neglecting local interactions makes it unable to satisfy geometric constraints (22.80% validity), redirecting model capacity toward fitting simpler global statistics rather than maintaining local physical realism.

In summary, these results demonstrate that DMFlow can be easily adapted to multi-component disordered systems with minimal architectural changes, and the primary challenge lies in the inherent complexity of the data distribution rather than the generative framework itself.

