# OpenReview forum: "DMFlow: Disordered Materials Generation by Flow Matching"
_ICLR.cc/2026/Conference — Submitted to ICLR 2026_

### Official Review · Reviewer_W2VS · 2025-10-30

**Soundness:** 3
**Presentation:** 3
**Contribution:** 2
**Rating:** 6
**Confidence:** 4

**Summary:**

DMFlow proposes an architecture specifically for disordered materials and curates multiple subsets of the Crystallography Open Database (COD) with disordered organic materials. Two types of disorder are considered: substitutional and positional disorder. The architecture includes a spherical geometric embedding of the substitutional disorder, positional disorder is a weighted contribution from two positions, and a voting technique for establishing disorder during inference.

**Strengths:**

- Interesting (novel) subject area given the growing interest in materials in deep learning. It also begins to attack real problems in materials science rather than the simplified "pure crystal" case.
- The methods (manifolds, voting) are well motived for the application and provide both anecdotal and numerical improvements. Stronger results for substitutional and positional disorder. Ablations against relevant alternative methods.

**Weaknesses:**

- The choice to focus on only binary disorder seems tailored to the dataset and not a strong representation of what disorder is possible. I would think a probability distribution over space would be a better representation of disordered atoms.
- I was surprised that the voting scheme did not seem to make major improvements over FlowMM-Prob in the substitutionally disordered (SD) case.

**Questions:**

- While the voting scheme is clever, it is ultimately a discretization. Is there any possibility for retaining a probabilisitic view of both substitutional and positional disorder and evaluating that?
- What are the limitations of using only two points of disorder?
- In non-disordered crystal generation we compute thermodynamic stability, is there a similar computation you can do for this case? Free energy?

---

> ### Author Response · Authors · 2025-11-21
> **Response to Reviewer W2VS – Part 1 / 2**
>
> ### **Summary**
> We sincerely thank the reviewer for the insightful comments. In our response, we **have addressed all points** as follows:  **(1) Extension to Higher-Order PD Crystals:** We extend our framework to Higher-Order PD to demonstrate its capacity for learning general spatial distributions beyond the binary setting; **(2) Clarification on Voting Scheme:** We clarify the specific applicability of the voting scheme; **(3) Rationale for Discretization:** We discuss the necessity of discretization over direct probabilistic evaluation; and **(4) Challenges in Stability Evaluation:** We explain the methodological barriers that prevent standard thermodynamic stability tools from being directly applied to probabilistic disordered structures.
>
> &nbsp;
>
> ### **Detailed Responses**
> > **W1: The choice to focus on only binary disorder seems tailored to the dataset and not a strong representation of what disorder is possible. I would think a probability distribution over space would be a better representation of disordered atoms.**
>
> We fully agree with your perspective that a probability distribution over space represents a more fundamental and accurate description of disordered atoms. Our initial focus on binary disorder was a **data-driven choice**, as the vast majority of structures in the COD are recorded as binary split sites, providing a rigorous baseline for this new task. To demonstrate the framework's capacity to learn more general spatial distributions, we have extended DMFlow to handle **Higher-Order PD Crystals (detailed in Appendix E)** from three aspects:
>
> 1. **Methodology.** We generalize the model by extending the fractional coordinates into a multi-channel matrix formulation and the disorder weights into a higher-dimensional categorical distribution. Crucially, we also adapt the GNN's interaction mechanism: rather than fixed concatenation, we employ a weighted summation of edge features across all potential split-site pairs. This ensures the model can flexibly aggregate geometric interactions weighted by their joint probabilities, maintaining permutation invariance regardless of the PD order. These techniques allow the flow matching model to jointly generate multiple potential split positions and their corresponding weights within a unified high-dimensional manifold, requiring minimal architectural changes.
>
> 2. **Dataset (COD-SHPD).** To support this, we construct a new dataset, COD-SHPD, by relaxing the atom count limit to 80 and explicitly including complex multi-site cases (at most 5 split positions). This resulted in a comprehensive dataset of 14,593 structures (including 5,497 with higher-order PD), providing a robust basis for training and evaluation.
>
> 3. **Empirical Verification.** We evaluate this extended model on COD-SHPD. The results (**Table R8**) confirm the method's applicability to broader disordered classes. Despite the significantly increased complexity, the model **achieves a Compositional Validity of 63.46% and Structural Validity of 48.16%, while maintaining over 97% in both Recall and Precision.** These findings confirm that DMFlow effectively captures the diversity of multi-component disordered structures and is robust beyond the binary setting.  Furthermore, we conduct ablation studies in this generalized setting. The results show that removing the Simplex Constraint causes a drastic drop in compositional validity (to 32.99%), and removing the Multi-Interaction modeling harms structural validity (to 22.80%). This reconfirms that our specialized architectural designs are essential for robustness beyond the binary setting. Although removing multi-interaction improves property metrics, we hypothesize that neglecting local interactions makes the model unable to meet necessary geometric constraints, shifting its capacity toward fitting simple global statistics like density rather than maintaining local physical realism.
>
> These results confirm that DMFlow is not limited to binary cases and can effectively approximate complex spatial probability distributions.
>
> &nbsp;
>
> **Table R8: DNG Results on COD-SHPD.**
> |Simplex|Multi-Interact.|Struc.|Comp.|Recall|Precision|Wdist($\rho$)|Wdist($N_{el}$)|
> |:---:|:---:|:---:|:---:|:---:|:---:|:---:|:---:|
> |✘|✔|25.61|32.99|68.85|76.44|1.939|0.898|
> |✔|✘|22.80|63.10|67.96|95.95|**0.452**|**0.713**|
> |✔|✔|**48.16**|**63.46**|**97.13**|**97.10**|0.731|1.146|
>
> &nbsp;
>
> > **W2:  I was surprised that the voting scheme did not seem to make major improvements over FlowMM-Prob in the substitutionally disordered (SD) case.**
>
> We appreciate the opportunity to clarify the experimental setup.  We clarify that the voting scheme **is not applicable to the CSP task** for SD, as the disorder weights are provided as fixed inputs rather than generated predictions, which explains why there is no performance difference attributed to it. The voting scheme is exclusively used in the DNG task, where disorder weights are generated from scratch and require discretization.

---

> ### Author Response · Authors · 2025-11-21
> **Response to Reviewer W2VS – Part 2 / 2**
>
> > **Q1: While the voting scheme is clever, it is ultimately a discretization. Is there any possibility for retaining a probabilisitic view of both substitutional and positional disorder and evaluating that?**
>
> Thanks for this crucial question. We agree that, in principle, retaining a fully probabilistic characterization is theoretically appealing. However, we employ discretization for two critical practical reasons:
>
> 1. **Physical Sparsity vs. Numerical Noise.** While our model operates on a high-dimensional simplex (up to 100 elements), **physical disorder is inherently sparse**: a single site typically contains a mixture of only 2-3 distinct species, not a diffuse "smear" of all elements. Raw probability vectors often contain **non-zero numerical noise** across many dimensions. Interpreting these tiny values as physical realities would be chemically incorrect. Therefore, discretization is necessary to filter this noise and recover the sparse, physically meaningful signal (i.e., the specific mixing components).
>
> 2. **Incompatibility with Standard Metrics.** Current standard metrics for materials validity are defined based on discrete assignments. For instance, Charge Neutrality checks require specific oxidation states derived from concrete element types. These metrics are mathematically undefined for a "diffuse probability mass."
>
> For these reasons, while our model generates a probabilistic representation, we must map it to a discrete crystal structure to perform a meaningful and standardized evaluation. We agree that developing new metrics capable of directly assessing probabilistic states is an exciting direction for future work.
>
>
> > **Q2: What are the limitations of using only two points of disorder?**
>
> We thank the reviewer for raising this insightful point. As discussed in our detailed response to **W1**, the focus on binary disorder was primarily a **data-driven choice** rather than a fundamental limitation of our generative framework.
>
> To address the limitation of this scope, we have successfully extended the framework to **Higher-Order PD Crystals (up to 5 split positions)**. By employing a multi-channel formulation, the extended model achieves robust performance on the newly constructed COD-SHPD benchmark (e.g., 63.46% Composional Validity and 97% Coverage). These results confirm that DMFlow is not restricted to the binary setting and can effectively generalize to more complex disordered systems.
>
> > **Q3: In non-disordered crystal generation we compute thermodynamic stability, is there a similar computation you can do for this case? Free energy?**
>
> Thank you for this forward-looking question. We agree that stability assessment is crucial for materials discovery, yet it remains underexplored for disordered crystals.
>
> For ordered crystals, stability is typically assessed using Ehull, computed either via DFT (e.g., VASP [2]) or ML potentials (e.g., MACE [3]). However, extending this evaluation to disordered crystals is fundamentally challenging. Unlike ordered structures, where the 0 K potential energy often serves as a reliable proxy, disordered materials require sampling from an enormous ensemble of atomic configurations (e.g., via Special Quasirandom Structures or Cluster Expansion [4,5]) to estimate thermodynamic stability. Current DFT and ML force field models cannot directly handle probabilistic occupancies or multi-position site vectors, as they require deterministic atomic configurations. This difficulty makes high-throughput stability evaluation for generative models of disordered crystals computationally prohibitive.
>
> These limitations underscore a broader gap in both the representation of disordered structures and the tools for their physical evaluation. Our work offers a unified formulation and a generative framework that treat ordered and disordered crystals consistently, which we hope will support future advances in both modeling and physics-based evaluation.
>
> &nbsp;
>
> [2] Hafner, Jürgen. "Ab‐initio simulations of materials using VASP: Density‐functional theory and beyond." Journal of computational chemistry 29.13 (2008): 2044-2078.
>
> [3] Batatia, Ilyes, et al. "MACE: Higher order equivariant message passing neural networks for fast and accurate force fields." Advances in neural information processing systems 35 (2022): 11423-11436.
>
> [4] van de Walle, Axel, et al. "Efficient stochastic generation of special quasirandom structures." Calphad 42 (2013): 13-18.
>
> [5] Wu, Qu, et al. "Cluster expansion method and its application in computational materials science." Computational Materials Science 125 (2016): 243-254.

---

> ### Comment · Reviewer_W2VS · 2025-11-25
> **glad to see new dataset**
>
> Thank you for your detailed reply. I appreciate the extension of the model (and data) to include more disorder. I see this as a very positive inclusion.
>
> I'm not quite as sure about the probability arguments w.r.t. "smearing" density. Although, I do see why their choices were motived by what appeared in the data. Perhaps the days of a probabilistic representation of atomic position are not here yet.
>
> Also the noise in the atom types strikes me as just as much of a problem as identifying atomic positions via a probabilistic method when the target measure is dirac. (i.e. single points.)
>
> Anyhow, these answers were fine. The addition of the data makes me give one more point.

---

> > ### Author Response · Authors · 2025-11-26
> > **Sincere Thanks for the Feedback**
> >
> > We sincerely thank the reviewer for the positive feedback and for acknowledging the value of our model extension and the new dataset. We are greatly encouraged by your decision to raise the score.
> >
> > We also fully share your vision regarding the future of probabilistic representations. While current crystallographic conventions largely rely on discrete split sites, we agree that moving towards continuous spatial probability distributions represents a promising frontier for describing atomic disorder.  Your insight on the parallel between atom type noise and the challenge of identifying specific "Dirac" positions from probabilistic outputs highlights a central theme in this field: the tension between learning smooth probabilistic measures and recovering discrete physical realities. We agree that this remains a fundamental challenge for the broader community, and we look forward to future developments that might bridge this gap.
> >
> > Thank you again for the stimulating discussion that has helped refine the positioning of our work.

---

### Official Review · Reviewer_imvy · 2025-10-31

**Soundness:** 2
**Presentation:** 3
**Contribution:** 2
**Rating:** 4
**Confidence:** 4

**Summary:**

This paper proposes DMFlow, the first deep generative framework for disordered crystals. It introduces a representation that jointly models ordered, substitutionally disordered (SD), and positionally disordered (PD) structures. Built on Riemannian flow matching with spherical reparameterization, DMFlow generate lattices, coordinates, and disorder weights simultaneously. A specialized graph neural network (GNN) captures crystallographic symmetries and multi-positional interactions. The authors also build the first benchmark dataset for disordered crystals, and experiments show that DMFlow outperforms existing models on both crystal structure prediction and de novo generation tasks.

**Strengths:**

* The paper tackles an **underexplored but relevant problem** in generative modeling—extending flow-based methods to **disordered crystal structures**.
* The work contributes a **new benchmark dataset** for disordered materials, which could be a useful resource for future research.

**Weaknesses:**

* **Relatively marginal improvement over baselines:**
  The experimental results show only small performance gains compared to **FlowMM**, suggesting that the proposed method offers limited novelty or practical advancement from a machine learning perspective. I wonder the performance compared to more recent crystal generation methods e.g. CrysBFN [1] and MatterGen [2]

* **Relatively minor machine learning contribution**
  The paper's contribution to the core machine learning field is relatively minor. It mainly adapts existing **flow matching frameworks** without introducing new algorithms or theoretical ML insights. The transition from ordered to disordered crystal generation appears straightforward, as the model only handles **binary positional disorder (two possible positions)** and adds several representation variables.

* **Insufficient ablation studies:**
  The paper lacks comprehensive analyses to justify the necessity of each proposed component. For instance, it would be valuable to compare the **spherical reparameterization** with a simpler flow matching setup to better demonstrate its actual effectiveness.

[1] Wu H, Song Y, Gong J, et al. A Periodic Bayesian Flow for Material Generation[C]//ICLR. 2025.

[2] Zeni, C., Pinsler, R., Zügner, D., Fowler, A., Horton, M., Fu, X., ... & Xie, T. (2025). A generative model for inorganic materials design. Nature, 639(8055), 624-632.

**Questions:**

1. The evaluation protocol used for DMFlow appears to be largely similar to that of CDVAE and other models designed for ordered materials. Given that disordered crystals have inherently different structural characteristics, it is unclear whether these evaluation metrics are fully appropriate. Could the authors justify or validate the suitability of applying ordered-material metrics to disordered systems?
2. The paper mentions that vanilla flow matching on the simplex can lead to numerical instability, which motivates the use of spherical reparameterization. Could the authors elaborate on the underlying causes of this instability and explain why the spherical mapping effectively mitigates it?

---

> ### Author Response · Authors · 2025-11-21
> **Response to Reviewer imvy – Part 1 / 4**
>
> ### **Summary**
> We sincerely thank the reviewer for the constructive feedback. In our response, we **address all comments** as follows: **(1) Validation of Performance Gains:** We analyze the performance gains and provided comparison with new baselines MatterGen; **(2) Clarification of ML Contributions:** We highlight the algorithmic novelty of our geometry-aware framework and task novelty; **(3) Expanded Ablation Studies:** We reinforce the necessity of the spherical constraint against Euclidean baselines and added new analyses on the discretization scheme; **(4) Justification of Evaluation Metrics:** We detail the rigorous adaptation of standard protocols for disordered systems; **(5) Mathematical Derivation:** We provide the theoretical background for the spherical reparameterization, explaining how it effectively resolves the numerical instabilities.
>
>  &nbsp;
> ### **Detailed Responses**
> > **W1: Relatively marginal improvement over baselines: The experimental results show only small performance gains compared to FlowMM, suggesting that the proposed method offers limited novelty or practical advancement from a machine learning perspective. I wonder the performance compared to more recent crystal generation methods e.g. CrysBFN [1] and MatterGen [2]**
>
> Thanks for the insightful comment. We appreciate the opportunity to clarify our contributions and compare against broader baselines.
>
> 1. **Clarifying the Baseline and Novelty.** First, we wish to emphasize that existing models like FlowMM are natively designed for ordered crystals and cannot handle disorder. The "baseline" in our paper is not an out-of-the-box model but a **significantly adapted version** we engineered (replacing embeddings for continuous SD vectors, splitting PD sites, etc.) to enable a fair comparison. Our contribution, DMFlow, goes beyond adaptation. It introduces a **simplex-constrained Riemannian flow matching** framework and a specialized **PD-aware GNN** to fundamentally solve the mathematical incompatibilities of disorder generation. From a machine learning perspective, modeling generative flows on constrained manifolds (Simplex/Sphere) to ensure physical validity is a non-trivial and meaningful advancement over standard Euclidean approaches.
>
> 2. **Analysis of Performance Gains.** Regarding performance, we would like to clarify that in the high-precision regime of CSP, the observed gains represent meaningful improvements in geometric fidelity:
>
>    +  **Quantitative Improvements:** DMFlow achieves substantial gains on MR and competitive RMSE across benchmarks, particularly on mixed-disorder (SPD) datasets. On COD-SPD-20, RMSE drops from 0.0724 to 0.0533 ( ~ 26% relative improvement) alongside improved Match Rate. Similarly, on the challenging COD-SPD-50, RMSE reduces from 0.0887 to 0.0725 ( ~ 18% relative improvement).
>
>    +  **Breaking the MR-RMSE Trade-off:** Crucially, this simultaneous improvement breaks the typical trade-off where a higher Match Rate inflates the average error by including "borderline" candidates. The fact that DMFlow reduces error while recovering more structures indicates a fundamental enhancement: it not only retrieves more ground-truth targets but predicts their atomic coordinates with significantly higher precision.
>
>    +  **Robustness in Complex Settings:** DMFlow exhibits significantly better stability than baselines when transitioning from SD to the complex SPD task. This confirms that our unified representation effectively captures the physical coupling between split positions, offering practical advantages that the heuristic site-splitting baseline fails to achieve.
>
> 3. **Comparison with New Baselines.** We thank you for suggesting CrysBFN and MatterGen. They are excellent recent works on ordered crystal generation, and we have added them to our related work discussion. To address your query, we evaluate MatterGen as follows:
>
>    +  **Implementation Details:** We implement and adapt **MatterGen** to the disordered CSP task using the same protocols (Prob/Sample/Max variants) based on its official codebase and comparable model parameters (Hidden_dim=384, Layers=2, Size=14.2M). For fairness, the model was trained from scratch on our datasets rather than using pre-trained weights, ensuring a direct comparison of architectural capabilities.
>
>    +  **Results Analysis:** The results (see **Table R6**) show that while MatterGen achieves a competitive RMSE, **it exhibits a significantly inferior Match Rate**. Unlike DMFlow, MatterGen fails to improve both metrics simultaneously, indicating it suffers from the selection trade-off discussed above.  Conventionally, a higher Match Rate is favored as it signifies a higher success rate in recovering ground-truth structures. Conversely, a low RMSE derived from a heavily truncated matched set is of limited practical utility, as it indicates the model fails to align a significant portion of the dataset.

---

> ### Author Response · Authors · 2025-11-21
> **Response to Reviewer imvy – Part 2 / 4**
>
> In a nutshell, this new comparison confirms that simply applying powerful ordered-crystal models is insufficient for disordered systems. Specialized architectural designs, such as modeling disorder weights on the simplex, are essential for achieving both high coverage and high precision.
>
> &nbsp;
>
> **Table R6: Comparison of CSP Results between MatterGen and DMFlow.**
> |Model|COD-SD-20 / MR(%)↑|COD-SD-20 / RMSE↓|COD-SPD-20 / MR(%)↑|COD-SPD-20 / RMSE↓|COD-SD-50 / MR(%)↑|COD-SD-50 / RMSE↓|COD-SPD-50 / MR(%)↑|COD-SPD-50 / RMSE↓|
> |---|:---:|:---:|:---:|:---:|:---:|:---:|:---:|:---:|
> |MatterGen-Sample|37.65|0.0585|32.73|0.0801|23.73|0.0643|19.23|0.0612|
> |MatterGen-Max|50.26|**0.0420**|49.83|0.0487|30.21|0.0456|20.85|**0.0488**|
> |MatterGen-Prob|53.87|0.0434|51.36|**0.0273**|37.14|**0.0434**|26.97|0.0517|
> |DMFlow|**70.57**|0.0738|**65.30**|0.0533|**49.12**|0.0681|**45.87**|0.0725|
>
> &nbsp;
>
> > **W2: Relatively minor machine learning contribution The paper's contribution to the core machine learning field is relatively minor. It mainly adapts existing flow matching frameworks without introducing new algorithms or theoretical ML insights. The transition from ordered to disordered crystal generation appears straightforward, as the model only handles binary positional disorder (two possible positions) and adds several representation variables.**
>
> Thanks for this comment. However, we respectfully disagree that the contribution is minor or that the transition from ordered to disordered systems is straightforward. We address this in three points:
>
> 1. **Establishing a New Benchmark (Zero-to-One Contribution).** Generative modeling for disordered crystals is a **new and underexplored task** that fundamentally differs from the ordered case due to the need to handle probabilistic occupancies and geometric constraints. This transition is far from trivial: it required us to **construct the first dedicated datasets** (COD-SD/SPD), design new evaluation protocols for probabilistic structures, and rigorously adapt existing baselines that were otherwise inapplicable. We believe establishing this complete benchmark ecosystem is a contribution to the ML-for-Science community.
>
> 2. **Algorithmic Novelty: Geometry-Aware Modeling.** From a modeling perspective, our contribution **goes far beyond merely "adding representation variables".** Standard Euclidean models cannot naturally handle the mathematical and physical constraints of disordered systems. DMFlow introduces specific algorithmic interventions to address this:
>
>     +  **Manifold Constraint Satisfaction:** Disordered weights reside on a probability simplex, not in Euclidean space. Naive Euclidean modeling (as shown in our "w/o Simplex" ablation) violates these constraints and fails to capture the sparse nature of physical occupancy. We introduced **Riemannian Flow Matching on the Sphere** (via spherical reparameterization) to rigorously solve the numerical instability of the Fisher-Rao metric near boundaries, ensuring the generative process respects the underlying geometry.
>     + **Unified Relational Modeling:** A key ML challenge is modeling the "one-to-many" mapping where a single site splits into coupled positions. Standard GNNs cannot process this. Instead of using heuristic site-splitting (which breaks semantic links), we designed a specialized **Unified GNN** with a full-pairs interaction mechanism. This allows the model to learn joint probability distributions over split sites explicitly, a capability absent in existing architectures.
>
> 3. **Generalization to Higher-Order PD Crystals.** Regarding the scope, our initial focus on binary PD was strictly **data-driven**, reflecting the dominant modality in the COD database. However, to demonstrate the framework's generality, we have extended the model to **Higher-Order PD** in Appendix E.
>
>     + **Methodology:** We generalize the model by extending the fractional coordinates into a multi-channel matrix formulation and the disorder weights into a higher-dimensional categorical distribution. Crucially, we also adapt the GNN's interaction mechanism: rather than fixed concatenation, we employ a weighted summation of edge features across all potential split-site pairs. This ensures the model can flexibly aggregate geometric interactions weighted by their joint probabilities, maintaining permutation invariance regardless of the PD order. These techniques allow the flow matching model to jointly generate multiple potential split positions and their corresponding weights within a unified high-dimensional manifold, requiring minimal architectural changes.
>
>     + **Dataset (COD-SHPD):** To support this, we construct a new dataset, COD-SHPD, by relaxing the atom count limit to 80 and explicitly including complex multi-site cases (at most 5 split positions). This resulted in a comprehensive dataset of 14,593 structures (including 5,497 with higher-order PD), providing a robust basis for training and evaluation.

---

> ### Author Response · Authors · 2025-11-21
> **Response to Reviewer imvy – Part 3 / 4**
>
> +  **Empirical Verification:** We evaluate this extended model on COD-SHPD. The results (**Table R7**) confirm the method's applicability to broader disordered classes. Despite the significantly increased complexity, the model **achieves a Compositional Validity of 63.46% and Structural Validity of 48.16%, while maintaining over 97% in both Recall and Precision.** These findings confirm that DMFlow effectively captures the diversity of multi-component disordered structures and is robust beyond the binary setting.  Furthermore, we conduct ablation studies in this generalized setting. The results show that removing the Simplex Constraint causes a drop in compositional validity (to 32.99%), and removing the Multi-Interaction modeling harms structural validity (to 22.80%). This reconfirms that our specialized architectural designs are essential for robustness beyond the binary setting. Although removing multi-interaction improves property metrics, we hypothesize that neglecting local interactions makes the model unable to meet necessary geometric constraints, shifting its capacity toward fitting simple global statistics like density rather than maintaining local physical realism.
>
> In summary, this work formalizes the task of disordered crystal generation, provides the first open-source benchmarks, and establishes a rigorous geometry-aware framework validated against adapted baselines. While our current results represent a first step rather than a final solution, we hope to inspire broader community engagement in accelerating the AI-driven discovery of this vital class of materials.
>
> &nbsp;
>
> **Table R7: DNG Results on COD-SHPD.**
> |Simplex|Multi-Interact.|Struc.|Comp.|Recall|Precision|Wdist($\rho$)|Wdist($N_{el}$)|
> |:---:|:---:|:---:|:---:|:---:|:---:|:---:|:---:|
> |✘|✔|25.61|32.99|68.85|76.44|1.939|0.898|
> |✔|✘|22.80|63.10|67.96|95.95|**0.452**|**0.713**|
> |✔|✔|**48.16**|**63.46**|**97.13**|**97.10**|0.731|1.146|
>
> &nbsp;
>
> > **W3: Insufficient ablation studies: The paper lacks comprehensive analyses to justify the necessity of each proposed component. For instance, it would be valuable to compare the spherical reparameterization with a simpler flow matching setup to better demonstrate its actual effectiveness.**
>
> We thank the reviewer for emphasizing the importance of rigorous component analysis. We would like to clarify that the comparison you requested is already covered in our experiments, and we have further expanded our ablation in the revision.
>
> 1. **Spherical Reparameterization vs. Simple Flow Matching.** We apologize if our initial presentation was not sufficiently clear, but the comparison you suggested is actually already presented in Table 2 as the "w/o Simplex" variant.  This variant represents the standard, unconstrained Euclidean flow matching with post-hoc normalization. The results demonstrate that this simpler setup fails to generate valid compositions, causing a drastic drop in Compositional Validity (e.g., from 69.06% to 29.63% on COD-SD-20) compared to our proposed Spherical Reparameterization. This confirms that the simplex constraint is a necessity for learning physically meaningful disorder weights.
> 2. **Additional Ablations on Discretization.** To further address your concern regarding comprehensive analysis, we have added a new ablation study on the Discretization/Voting Scheme in **Fig. 6 in our revised paper**:
>     +  **Individual Heuristics vs. Ensemble Voting:** The results reveal that while single heuristics can perform well on specific metrics, they suffer from distinct trade-offs. For example, Threshold and Top-k tend to produce poor Wdist($\rho$), whereas Entropy-based and Adaptive-threshold variants yield worse Wdist($N_{el}$). In contrast, our ensemble voting scheme effectively mitigates these individual biases. It achieves **the best compositional validity** and matches the strongest individual heuristics on structural validity, precision, and recall. By aggregating diverse decision rules, the ensemble delivers the most **balanced and reliable performance** across all metrics, justifying its design over any single heuristic.
>     +   **Sensitivity to Thresholds:** To address your concern about robustness, we perform a sensitivity analysis by perturbing these thresholds.  The results demonstrate high stability, structural validity and coverage metrics remaining essentially unchanged across all settings.  While we observe minor fluctuations in property distances, the overall performance remains stable. These findings confirm that our voting scheme is **robust to moderate threshold perturbations** and that the hyperparameters used in the paper are located within a stable operating region.
>
> Ultimately, the combination of our existing ablations (validating the spherical manifold design) and the new detailed analysis of the discretization module (validating the voting scheme) provides a comprehensive justification for the necessity of all key architectural components in DMFlow.

---

> ### Author Response · Authors · 2025-11-21
> **Response to Reviewer imvy – Part 4 / 4**
>
> > **Q1:  The evaluation protocol used for DMFlow appears to be largely similar to that of CDVAE and other models designed for ordered materials. Given that disordered crystals have inherently different structural characteristics, it is unclear whether these evaluation metrics are fully appropriate. Could the authors justify or validate the suitability of applying ordered-material metrics to disordered systems?**
>
> We thank the reviewer for this crucial question. We agree that disordered crystals possess unique characteristics that require careful handling. While we adopt the standard metric categories to ensure consistency with the broader literature, we clarify that these metrics are **not applied blindly**. As detailed in **Appendix D.3**, we have rigorously adapted the implementation of each metric to ensure its mathematical and physical validity in the probabilistic setting.
>
> 1. For CSP, the evaluation relies on Match Rate and RMSE, computed using Pymatgen's `StructureMatcher[1]`. This tool operates on the fundamental geometry (lattice and coordinates) to determine if two structures are crystallographically equivalent within a tolerance. Since the goal of CSP is to recover the correct spatial arrangement of sites, these geometric metrics remain fully appropriate for assessing whether the generated structure aligns with the ground truth, regardless of site occupancy types.
>
> 2. For DNG, standard metrics required specific modifications to handle fractional weights.  We introduced specific generalizations to handle probabilistic weights. Compositional validity was extended to verify charge neutrality using fractional oxidation states. For coverage metrics, which rely on structural fingerprints incompatible with disorder, we implemented a stochastic ensemble strategy: averaging fingerprints from multiple ordered realizations to create a robust and unbiased representation. Similarly, property statistics were adapted to sum weighted contributions rather than integers. These tailored implementations ensure the metrics meaningfully reflect the nature of disordered systems.
>
> In conclusion, while the high-level evaluation framework parallels that of ordered materials, our specific implementations have been carefully tailored to reflect the probabilistic nature of disordered systems.
>
> [1] Ong, Shyue Ping, et al. "Python Materials Genomics (pymatgen): A robust, open-source python library for materials analysis." Computational Materials Science 68 (2013): 314-319.
>
> &nbsp;
>
> >  **Q2: The paper mentions that vanilla flow matching on the simplex can lead to numerical instability, which motivates the use of spherical reparameterization. Could the authors elaborate on the underlying causes of this instability and explain why the spherical mapping effectively mitigates it?**
>
> Thank you for this insightful question. We are happy to clarify the motivation behind the spherical reparameterization and to further supplement the discussion in Section 4.2.
>
> The numerical instability of vanilla flow matching on the simplex mainly arises from the Fisher–Rao geometry of categorical distributions. On the simplex, the Riemannian inner product $\langle u, v\rangle_\mu=\sum_i \frac{u_i v_i}{\mu_i}$ becomes ill-conditioned near the boundary ($\mu_i\to 0$), **causing both the metric and the logarithm/exponential maps to blow up.  In other words,  the geodesics, which flow matching crucially depends on, will be numerically unstable or even undefined.**
>
> To address this, we reparameterized via the diffeomorphism $\pi:\mu_i \to x_i=\sqrt{\mu_i} $,  which maps the simplex to the positive orthant of the sphere. On the sphere, the inner product is simply the Euclidean dot product: $\langle u, v\rangle_\mathbb{S}=\sum_i u_i v_i$, which remains finite and stable even when $\mu_i=0$. As a result, geodesics and vector fields can be computed in closed form without singularities, and training becomes numerically stable.
>
> In essence, the spherical mapping removes the $1/\mu_i$ singularity while preserving the correct geometry, which is why it effectively mitigates the instability of direct flow matching on the simplex.

---

> ### Author Response · Authors · 2025-11-26
> **Kind Reminder on Rebuttal Deadline**
>
> Dear Reviewer imvy,
>
> We truly appreciate your time and expertise on reviewing our paper. **We’ve already revised the manuscript in light of the feedback received so far. As the rebuttal deadline is fast approaching**, we want to kindly check **whether you might have any additional comments or concerns** for us to address.
>
> We’d be grateful for any feedback you could share before the deadline.  Thank you again for your support!
>
> Sincerely,
>
> Authors

---

### Official Review · Reviewer_XWs2 · 2025-10-31

**Soundness:** 3
**Presentation:** 3
**Contribution:** 3
**Rating:** 6
**Confidence:** 3

**Summary:**

This paper tackles the generation of disordered materials, extending recent advances in generative models that have focused exclusively on ordered crystalline materials where atom types and positions are fully determined. The proposed approach handles two types of disorder: Substitutional Disorder (SD), where multiple atom types can occupy the same site, and Positional Disorder (PD), where atoms can occupy multiple positions. For SD, they replace one-hot atom type encodings with occupancy vectors on the simplex. For PD, they focus on the binary case, introducing weight vectors and additional fractional coordinates per atom. To process these new site-level features, they develop a novel GNN architecture and a sampling procedure that uses majority voting across five heuristics based on occupancy and weight logits. The authors introduce new datasets for disordered materials modeling and demonstrate that their approach successfully generates such materials, representing an important first step in this domain.

**Strengths:**

The paper is quite strong in terms of novelty, as it is the first work I am aware of that applies diffusion models to disordered materials. It is clearly written, and the planned release of the datasets represents an important step toward enabling more generative models for this type of problem.

**Weaknesses:**

I believe the paper would benefit from a more detailed discussion of the design of the CSP experiment (see questions below). There are also some design choices that are not fully ablated, such as the use of five different heuristics for the majority vote during sampling. It would be interesting to see how each heuristic performs individually and whether all of them are necessary. The framework also relies heavily on manually set thresholds during sampling, and it is not entirely clear how the values reported in the appendix were selected or whether alternative values were considered. Additionally, it would be valuable to understand how small changes in these thresholds affect the final performance of the model.

**Questions:**

- I have one question regarding the CSP experiments. It might be a naive question, but why are you considering W, the weight over the two possible coordinates, to be given and observed? Shouldn’t the model be learning it as part of the task?
- A bit related to the previous question, in the case of CSP with SPD disorder, what are the main differences between FlowMM-Prob and DMFlow? Is it in the way the weights over the two fractional coordinates are encoded? How can the difference in performance between the two methods be explained? What prevented you from using the same conditioning mechanism in both models, given that in this case the model’s output consists only of the lattice parameters and the two sets of fractional coordinates?
- Are the five heuristics in the majority vote needed? How do results change if one consider a single heuristic or a subset of that? Additionally, how sensitive is the final performance to small variations in the threshold values?

---

> ### Author Response · Authors · 2025-11-21
> **Response to Reviewer XWs2 – Part 1 / 2**
>
> ### **Summary**
> We appreciate the reviewer’s careful reading and  the constructive suggestions that helped us improve the paper. In the following responses, **we have addressed all comments point-by-point.** Key clarifications and updates include: **(1) A comprehensive ablation of the discretization module**, validating the necessity of the ensemble voting scheme and demonstrating its sensitivity of thresholds; **(2) A clarification of the CSP task design**, justifying the rationale for observing disorder weights and explaining the advantages of our unified representation; **(3) A detailed discussion on baseline configurations**, highlighting the non-trivial adaptations required to extend ordered-crystal generative models to the disordered domain.
>
>  &nbsp;
> ### **Detailed Responses**
> > **W1: The CSP design lacks detail, key choices (five heuristics, sampling thresholds) are not ablated, and the sensitivity of these thresholds is unclear.**
>
> We thank the reviewer for this constructive suggestion. We agree that a deeper analysis of the discretization strategy strengthens the paper. We have conducted a comprehensive ablation study (**Fig. 6 of the revised paper**) to evaluate individual heuristics and the sensitivity of hyper-parameters.
>
> 1. **Individual Heuristics vs. Ensemble Voting.** The results reveal that while single heuristics can perform well on specific metrics, they suffer from distinct trade-offs. For example, Threshold and Top-k tend to produce poor Wdist($\rho$), whereas Entropy-based and Adaptive-threshold variants yield worse Wdist($N_{el}$). In contrast, our ensemble voting effectively mitigates these individual biases. It achieves **the best compositional validity** and matches the strongest individual heuristics on structural validity, precision, and recall. By aggregating diverse decision rules, the ensemble delivers the most **balanced and reliable performance** across all metrics, justifying its design over any single heuristic.
>
> 2. **Sensitivity to Thresholds.** To address your concern about robustness, we perform a sensitivity analysis by perturbing these thresholds. The results demonstrate high stability, structural validity and coverage metrics remaining essentially unchanged across all settings. While we observe minor fluctuations in property distances, the overall performance remains stable. These findings confirm that our voting scheme is **robust to moderate threshold perturbations** and that the hyper-parameters used in the paper are located within a stable operating region.
>
> In summary, these additional experiments validate that the ensemble design is essential for achieving balanced performance across conflicting metrics and exhibits low sensitivity to threshold variations.
>
> > **Q1: I have one question regarding the CSP experiments. It might be a naive question, but why are you considering W, the weight over the two possible coordinates, to be given and observed? Shouldn’t the model be learning it as part of the task?
> Thank you for this insightful question. It touches on the fundamental definition of the CSP task for disordered materials.**
>
> Thank you for this insightful question. It touches on the fundamental definition of the CSP task for disordered materials.
>
> 1. **The Essence of CSP: Solving Geometry for A Given State.** In standard CSP for ordered crystals, the input is the exact composition (e.g., $\text{Fe}_2\text{O}_3$), and the goal is to find the stable geometry. For disordered materials, the "composition" or thermodynamic state is more complex: it is defined not only by the chemical ratios (SD weights) but also by the spatial occupancy probabilities (PD weights,). Therefore, we defined the CSP task here as: **given a specific disordered configuration, determine the corresponding equilibrium geometry (Lattice and Coordinates).** Treating $W$ as an input constraint ensures that the model focuses on resolving the geometric realization of a specific, experimentally observed or theoretically defined disordered phase.
>
> 2. **Potential for Predicting Weights.** We fully agree with your point that learning weights is a highly interesting task. A **"Generalized CSP"** setting, where the model takes only the chemical elements as input and simultaneously predicts the optimal structure and the disorder weights, would be very valuable for applications like discovering optimal doping concentrations. While our current experiment focuses on geometric reconstruction under fixed constraints, our generative framework is flexible enough to support such tasks in future work.
>
> Overall, treating $W$ as an observed input is a deliberate design choice to rigorously define the CSP task as "solving for geometry given a specific thermodynamic state." This setup allows us to isolate and evaluate the model's geometric reconstruction capability. Meanwhile, the capacity to learn and predict these disorder weights from scratch is interesting and promising to design a task of "Generalized CSP".

---

> ### Author Response · Authors · 2025-11-21
> **Response to Reviewer XWs2 – Part 2 / 2**
>
> > **Q2: A bit related to the previous question, in the case of CSP with SPD disorder, what are the main differences between FlowMM-Prob and DMFlow? Is it in the way the weights over the two fractional coordinates are encoded? How can the difference in performance between the two methods be explained? What prevented you from using the same conditioning mechanism in both models, given that in this case the model’s output consists only of the lattice parameters and the two sets of fractional coordinates?**
>
> You are absolutely correct that the core difference lies in how Positional Disorder (PD) is encoded and processed. We address your questions in three parts:
>
> 1. **Encoding Strategy.** The fundamental difference is that FlowMM-Prob relies on a "site splitting" heuristic (detailed in Appendix D.2.1), where a single PD site is decomposed into two independent nodes to fit the standard GNN architecture. In contrast, **DMFlow employs a unified representation** (Section 4.3), treating the PD site as a single entity with multiple probabilistic coordinates.
>
> 2. **Performance Impact.** This difference directly explains the performance gap. By splitting the site, the baseline forces the model to treat coupled split positions as independent atoms, effectively severing their semantic link. DMFlow, however, preserves this connection and explicitly models the physical coupling via **joint probability interactions** (Eq. 13 & 14). This allows our model to capture the intrinsic correlations that the baseline ignores, leading to significantly higher structural precision.
>
> 3. **Why Different Conditioning Mechanisms.** We could not apply our unified conditioning to the baseline because standard GNN architectures (like the one in FlowMM) are fundamentally constrained by a "**one-node-one-coordinate**" paradigm. They inherently assume that each graph node corresponds to a unique spatial position. They lack the internal mechanisms to process a node associated with multiple spatial coordinates simultaneously. Consequently, "site splitting" was the only viable way to map disordered data into a format compatible with the baseline's architecture, effectively creating a "pseudo-ordered" input that fits its structural assumptions.
>
> Ultimately, the performance disparity resolves to a fundamental architectural mismatch: standard GNNs require "pseudo-ordered" inputs that dilute physical correlations, whereas DMFlow's unified design natively accommodates the probabilistic nature of disorder. This structural advantage allows our model to learn tighter geometric constraints, directly translating to the observed improvements in prediction accuracy.
>
> > **Q3: Are the five heuristics in the majority vote needed? How do results change if one consider a single heuristic or a subset of that? Additionally, how sensitive is the final performance to small variations in the threshold values?**
>
> Thank you for the follow-up question. As detailed in our response to **W1**, we have conducted a comprehensive ablation study to address these specific points.
>
> Regarding the heuristics, our evaluation confirms that no single method performs optimally across all metrics. The ensemble of five heuristics is indeed necessary to mitigate these individual biases, **delivering the most balanced and robust performance** compared to any single subset.
>
> Regarding threshold sensitivity, our perturbation analysis shows that structural validity and coverage metrics remain essentially unchanged under moderate variations. This confirms that **the final performance is robust** and not overly sensitive to small changes in threshold values.

---

> ### Author Response · Authors · 2025-11-26
> **Request for Further Comments Before Rebuttal Deadline**
>
> Dear Reviewer XWs2,
>
> Thank you sincerely for your careful consideration of our submission.  **With the rebuttal deadline drawing near, we’ve worked diligently to resolve all points**. To ensure we present the most complete and improved version possible, **we would be very grateful to know if you have any further feedback or questions for us.**
>
> Please don’t hesitate to share your thoughts, even brief comments would be extremely helpful. Thanks again for your time and thoughtful engagement with our work.
>
> Warm regards,
>
> Authors

---

### Official Review · Reviewer_y4YR · 2025-11-01

**Soundness:** 2
**Presentation:** 3
**Contribution:** 2
**Rating:** 2
**Confidence:** 3

**Summary:**

The paper aims to address the lack of generative models that natively handle disordered crystals—both substitutional (SD) and positional (PD)—and the absence of a public benchmark for this setting. It proposes DMFlow, a flow-matching framework with modules that tailored for disordered crystals. These modules include 1) a unified, site-centric representation for ordered/SD/PD crystals; 2) Riemannian flow matching on the simplex via spherical reparameterization to ensure valid disorder weights; 3) a symmetry-aware GNN for velocity prediction, and 4) a two-stage discretization (ordered-site detection + ensemble voting) to convert continuous weights into multi-hot assignments. The authors also build a COD-derived benchmark covering SD and mixed SD+PD (binary PD) with standard splits. On CSP, DMFlow matches FlowMM-Prob on pure SD (by design) but improves on mixed SPD. On DNG, ablations show necessity of the simplex constraint and multi-interaction modeling.

**Strengths:**

1. The paper proposes a unified representation for both ordered and disordered crystals.

2. On the CSP tasks, the proposed method outperforms the baseline methods on a SPD dataset - a mixed set combining both SD and PD structures.

3. The paper contributes a new dataset for benchmarking disordered crystal generation.

**Weaknesses:**

1. The scope of the paper is limited. The paper only considers binary PD while leaving generality to >2 positions per site untested. This may limit applicability to broader disordered classes.

2. In the CSP experiments, the proposed DMFlow only marginally outperforms the baseline FlowMM-Prob on binary PD cases.

3. In the DNG experiments, the authors do not compare against existing generative baselines. The authors argue that the existing methods can only generate one-hot element assignments. However, this rationale is not entirely compelling to me. On the top of my head, softmax or gumbel softmax could be adapted to yield fractional or multi-hot element assignments.

**Questions:**

The authors claim that existing methods can only generate one-hot element assignments which motivates the discretization module presented in Section 4.4. The discretization module seems to introduce many complex biases. Did the authors try to adapt softmax or gumbel softmax? if not, can the authors provide a justification for why these approaches are unsuitable?

---

> ### Author Response · Authors · 2025-11-21
> **Response to Reviewer y4YR – Part 1 / 3**
>
> ### **Summary**
> We sincerely thank the reviewer for the constructive comments, which have significantly strengthened our paper. **We provide point-by-point responses to all raised questions**. Key updates include: **(1) Extending to Higher-Order PD Crystals:** We generalize the framework to support Higher-Order PD with a new dataset. Results confirm the model's capacity to learn complex spatial distributions.  **(2) Validation of Performance Gains:** We clarify the intrinsic difficulty of the task and highlighted that DMFlow achieves chemically significant gains in geometric precision and superior robustness. **(3) Ablation on Softmax Normalization:** We add comparisons against a Softmax-based baseline. The poor performance of this variant conclusively validates the necessity of our intrinsic flow matching design. **(4) Ablation on Discretization:** We provide a detailed evaluation of the voting scheme.
>
> &nbsp;
> ### **Detailed Responses**
>
> > **W1:  The scope of the paper is limited. The paper only considers binary PD while leaving generality to >2 positions per site untested. This may limit applicability to broader disordered classes.**
>
> Thanks for this constructive comment. We respectfully wish to clarify our scope and generality in two parts:
> 1. **Foundational Nature of the Work.** This work aims to address the significant gap in the relatively underexplored domain of disordered crystal generation, rather than a limited study. This task is fundamentally more challenging than ordered generation, as it requires handling probabilistic occupancies and complex geometric constraints that standard models cannot address. DMFlow establishes the first unified framework capable of jointly generating SD and PD. **Our initial focus on binary PD was strictly data-driven:** it reflects the vast majority of entries in the COD database, serving as the most rigorous starting point to validate this new methodology on the dominant data modality.
>
> 2. **Extension to Higher-Order PD Crystals.** To demonstrate that our framework is not restricted to binary cases, we have extended DMFlow to handle Higher-Order PD, as detailed in Appendix E.
>
>     + **Methodology:** We generalize the model by extending the fractional coordinates into a multi-channel matrix formulation and the disorder weights into a higher-dimensional categorical distribution. Crucially, we also adapt the GNN's interaction mechanism: rather than fixed concatenation, we employ a weighted summation of edge features across all potential split-site pairs. This ensures the model can flexibly aggregate geometric interactions weighted by their joint probabilities, maintaining permutation invariance regardless of the PD order. These techniques allow the flow matching model to jointly generate multiple potential split positions and their corresponding weights within a unified high-dimensional manifold, requiring minimal architectural changes.
>
>     + **Dataset (COD-SHPD):** To support this, we construct a new dataset, COD-SHPD, by relaxing the atom count limit to 80 and explicitly including complex multi-site cases (at most 5 split positions). This results in a comprehensive dataset of 14,593 structures (including 5,497 with higher-order PD), providing a robust basis for training and evaluation.
>
>     + **Empirical Verification:** We evaluate this extended model on COD-SHPD. The results (**Table R1**) confirm the method's applicability to broader disordered classes. Despite the significantly increased complexity, the model **achieves a Compositional Validity of 63.46% and Structural Validity of 48.16%, while maintaining over 97% in both Recall and Precision.** These findings confirm that DMFlow effectively captures the diversity of multi-component disordered structures and is robust beyond the binary setting. Furthermore, we conduct ablation studies in this generalized setting. The results show that removing the Simplex Constraint causes a drastic drop in compositional validity (to 32.99%), and removing the Multi-Interaction modeling harms structural validity (to 22.80%). This reconfirms that our specialized architectural designs are essential for robustness beyond the binary setting. Although removing multi-interaction improves property metrics, we hypothesize that neglecting local interactions makes the model unable to meet necessary geometric constraints, shifting its capacity toward fitting simple global statistics like density rather than maintaining local physical realism.
>
> In short, these extensive new experiments demonstrate that DMFlow is a generalizable framework capable of handling complex, higher-order PD beyond the binary setting.
>
> &nbsp;
>
> **Table R1: DNG Results on COD-SHPD.**
>
> |Simplex|Multi-Interact.|Struc.|Comp.|Recall|Precision|Wdist($\rho$)|Wdist($N_{el}$)|
> |:---:|:---:|:---:|:---:|:---:|:---:|:---:|:---:|
> |✘|✔|25.61|32.99|68.85|76.44|1.939|0.898|
> |✔|✘|22.80|63.10|67.96|95.95|**0.452**|**0.713**|
> |✔|✔|**48.16**|**63.46**|**97.13**|**97.10**|0.731|1.146|

---

> ### Author Response · Authors · 2025-11-21
> **Response to Reviewer y4YR – Part 2 / 3**
>
> > **W2: In the CSP experiments, the proposed DMFlow only marginally outperforms the baseline FlowMM-Prob on binary PD cases.**
>
> We thank the reviewer for this observation. We address the concerns from two perspectives:
>
> 1. **Non-trivial Baseline Adaptation.** It is important to clarify that **FlowMM is natively designed for ordered crystals and cannot handle PD.** To enable a valid comparison, we significantly adapted it into FlowMM-Prob by manually splitting PD sites into independent nodes, without which the baseline would be inapplicable. Unlike this adapted baseline which treats split sites independently, DMFlow explicitly models their physical correlations via a unified framework. Specifically, it employs simplex-constrained flow matching to ensure physically valid disorder weights and a specialized GNN that aggregates multiple pairwise interactions to capture the complex geometry of split sites. This represents a theoretically grounded approach rather than a heuristic adaptation.
>
> 2. **Meaningful Performance Improvement.** Regarding CSP performance, DMFlow achieves consistent improvements on MR and competitive RMSE values across datasets, **particularly on mixed-disorder (SPD) benchmarks.** On COD-SPD-20, DMFlow reduces the RMSE from 0.0724 to 0.0533 ($\sim$26% relative improvement) while also improving the Match Rate (65.30% vs. 64.16%). Similarly, on the more challenging COD-SPD-50 dataset, it not only improves the Match Rate (from 44.00% to 45.87%) but, more notably, achieves a substantial gain in structural precision, reducing the RMSE from 0.0887 to 0.0725 ($\sim$ 18% relative improvement). Crucially, this simultaneous improvement breaks the **typical trade-off observed in structure prediction**, where a higher Match Rate usually inflates the average error (RMSE) by including more "borderline" candidates with larger deviations. The fact that DMFlow reduces error while recovering more structures indicates a fundamental enhancement in geometric fidelity: **it not only retrieves more ground-truth targets but predicts their atomic coordinates with significantly higher precision than the baseline.**
>
> Overall, the results underscore the gap between a heuristic baseline and our principled unified framework. Improvements in both MR and RMSE show that DMFlow provides a core advantage in capturing geometric fidelity, particularly for mixed-disorder systems.
>
> > **W3: In the DNG experiments, the authors do not compare against existing generative baselines. The authors argue that the existing methods can only generate one-hot element assignments. However, this rationale is not entirely compelling to me. On the top of my head, softmax or gumbel softmax could be adapted to yield fractional or multi-hot element assignments.**
>
> We appreciate this insightful comment. We understand the concern regarding the comparison with adapted baselines. We would like to address this in three key points:
>
> 1. **The “w/o Simplex” ablation is the adapted baseline.** We wish to clarify that the ablation labeled "w/o Simplex" in Table 2 serves as the baseline you are looking for (i.e., an adapted FlowMM). This configuration generates disorder weights via unconstrained flow matching in Euclidean space, then applies L1 normalization to enforce the sum-to-one constraint. Its poor performance (e.g., 29.63% compositional validity on COD-SD-20) demonstrates that simple Euclidean adaptation is fundamentally insufficient for this task.
>
> 2. **New Experiment: Adapted Baselines via Softmax.** To fully validate your specific suggestion, we implement a new baseline using Softmax normalization instead of L1 on the Euclidean outputs.
>
>     + **Necessity of Discretization:** We must emphasize that, like L1 normalization, Softmax only enforces the sum-to-one constraint. It yields dense probability vectors and does not address the critical problem of **determining assignment sparsity**, that is, how many elements should occupy a site. Raw probability vectors also **contain small numerical noise** across many dimensions, and interpreting these tiny values as real occupancies would be chemically incorrect. Thus, our discretization procedure (Section 4.4) remains essential for converting Softmax outputs into valid multi-hot assignments.
>
>     + **Experimental Results:** As shown in the **table R2–R5**, the Softmax baseline yields a Compositional Validity of only 22.69% on COD-SD-20, which is even lower than the L1 baseline (29.63%) and substantially outperformed by DMFlow (69.06%). We attribute this to the fact that Softmax tends to produce higher-entropy (flatter) distributions, making it harder for the discretization module to extract clear, sparse atomic assignments compared to the sharper distributions learned via Simplex flow.

---

> ### Author Response · Authors · 2025-11-21
> **Response to Reviewer y4YR – Part 3 / 3**
>
> 3. **Why Post-hoc Normalization Fails.** The results confirm that geometry matters during the generation process, not just at the output.
>
>     + **Geometric Distortion:** Flow matching relies on learning a vector field that follows a straight path between the prior and data. When modeling in Euclidean space and applying post-hoc normalization (L1 or Softmax), the "straight lines" in Euclidean space map to highly curved, non-geodesic trajectories when projected onto the simplex. This mismatch creates a complex, unnatural vector field that is difficult for the neural network to learn.
>
>     + **Intrinsic Advantage:** DMFlow succeeds by performing flow matching directly on the Riemannian manifold (via spherical reparameterization). This ensures that the generated trajectory naturally follows the geodesic (shortest path) on the manifold, respecting the probabilistic structure at every step of the generation, rather than forcing it only at the end.
>
> In conclusion, our experiments confirm that adapted baselines (whether via L1 or Softmax) yield suboptimal results, validating the necessity of DMFlow's intrinsic Riemannian formulation. Thank you again for prompting this valuable clarification.
>
> &nbsp;
>
> **Table R2: DNG Results on COD-SD-20.**
> |Prob.Modeling|Multi-Interact.|Struc.|Comp.|Recall|Precision|Wdist($\rho$)|Wdist($N_{el}$)|
> |:---:|:---:|:---:|:---:|:---:|:---:|:---:|:---:|
> |L1-Norm|N/A|78.28|29.63|97.89|91.00|0.782|2.294|
> |Softmax|N/A|78.94|22.69|97.54|63.07|2.588|1.260|
> |Simplex|N/A|**88.14**|**69.06**|**98.59**|**93.55**|**0.155**|**0.691**|
>
> &nbsp;
>
> **Table R3: DNG Results on COD-SPD-20.**
> |Prob.Modeling|Multi-Interact.|Struc.|Comp.|Recall|Precision|Wdist($\rho$)|Wdist($N_{el}$)|
> |:---:|:---:|:---:|:---:|:---:|:---:|:---:|:---:|
> |L1-Norm|✔|69.62|54.99|96.41|90.56|0.695|1.687|
> |Softmax|✔|69.57|56.34|96.09|89.54|1.236|1.990|
> |Simplex|✘|72.54|69.32|96.09|94.12|0.284|**0.776**|
> |Simplex|✔|**87.30**|**69.99**|**99.18**|**94.31**|**0.259**|0.825|
>
> &nbsp;
>
> **Table R4: DNG Results on COD-SD-50.**
> |Prob.Modeling|Multi-Interact.|Struc.|Comp.|Recall|Precision|Wdist($\rho$)|Wdist($N_{el}$)|
> |:---:|:---:|:---:|:---:|:---:|:---:|:---:|:---:|
> |L1-Norm|N/A|81.23|36.41|94.72|94.15|0.257|3.301|
> |Softmax|N/A|81.57|20.98|97.25|68.62|2.618|2.354|
> |Simplex|N/A|**90.84**|**68.57**|**97.30**|**94.75**|**0.151**|**1.583**|
>
> &nbsp;
>
> **Table R5: DNG Results on COD-SPD-50.**
> |Prob.Modeling|Multi-Interact.|Struc.|Comp.|Recall|Precision|Wdist($\rho$)|Wdist($N_{el}$)|
> |:---:|:---:|:---:|:---:|:---:|:---:|:---:|:---:|
> |L1-Norm|✔|53.73|52.72|94.63|92.56|1.232|1.762|
> |Softmax|✔|52.98|53.54|94.12|91.98|1.075|1.853|
> |Simplex|✘|76.93|64.48|97.19|93.54|0.240|**1.343**|
> |Simplex|✔|**86.02**|**68.15**|**97.87**|**96.10**|**0.233**|1.531|
>
> &nbsp;
>
> > **Q1: The authors claim that existing methods can only generate one-hot element assignments which motivates the discretization module presented in Section 4.4. The discretization module seems to introduce many complex biases. Did the authors try to adapt softmax or gumbel softmax? if not, can the authors provide a justification for why these approaches are unsuitable?**
>
> We thank the reviewer for this question, which allows us to clarify the specific role of our discretization module.
>
> 1. **Distinction between Generation and Determination.** We agree that Softmax or Gumbel-Softmax are standard methods for generating probability vectors. However, they only ensure the output sums to one and do not determine the sparsity (i.e., the specific number of elements occupying a site). Therefore, **a discretization step is unavoidable** to convert probabilities into discrete multi-hot assignments, regardless of whether the probabilities come from Softmax or our Simplex flow.
>
> 2. **Mitigating Bias via Ensembling.** Far from introducing bias, our ensemble voting scheme is explicitly designed to mitigate the sensitivity and biases of individual heuristics. As demonstrated in our new ablation study (**Fig. 6 in the revised paper**), single rules suffer from distinct trade-offs: for instance, rigid methods like TopK tend to degrade $\text{Wdist}(\rho)$, while adaptive thresholds often affect $\text{Wdist}(N_{el})$. By aggregating these diverse rules, the ensemble approach effectively cancels out individual failure modes, delivering the most balanced performance across all metrics and demonstrating high robustness to threshold perturbations.
>
> 3. **Empirical Validation.** As detailed in our response to W3, we empirically test the "Softmax + Discretization" baseline. It underperforms DMFlow. This confirms that while discretization is a necessary post-processing step, the superior performance of DMFlow stems from learning the correct geometry during the generation process.
>
> In summary, discretization is necessary for sparse atomic assignments. The ensemble scheme reduces heuristic bias, and DMFlow’s gains over the Softmax baseline show that its advantage lies in intrinsic geometric learning rather than post-processing.

---

> ### Author Response · Authors · 2025-11-26
> **Gentle Reminder: Feedback Needed Before Upcoming Deadline**
>
> Dear Reviewer y4YR,
>
> Thank you for your time and valuable comments on our submission. We’re writing to kindly note that **the rebuttal deadline is approaching soon. We have carefully addressed all concerns raised so far** and are eager to incorporate any additional feedback you might have.
>
> If you have further suggestions or questions, **we would greatly appreciate your feedback at your earliest convenience.** Thank you once more for your support and guidance!
>
> Best regards,
>
> Authors

---

### Author Response · Authors · 2025-12-02
**Summary of Rebuttal Updates and Reviewer Discussions**

Dear Area Chairs,

We would like to express our sincere appreciation for your additional efforts in managing this unprecedented situation following the OpenReview information leak. The goal of this note is to provide a brief, high-level summary to support an efficient evaluation of our rebuttal under the revised process. Specifically, we summarize (1) the contribution and novelty of our work; (2) the current status of reviewer–author discussions, and (3) the key concerns raised during review along with how we addressed them through clarifications and new experimental results.

---
### **Paper Summary: Contributions and Novelty**
This paper addresses the significant gap in generative modeling for **disordered crystals**, a vital class of materials (e.g., superconductors) that standard ordered-crystal models cannot handle. We introduce **DMFlow**, the first unified framework capable of jointly generating Substitutional and Positional Disorder (SD and PD).

1. **Methodological Innovation:** We propose a **Riemannian flow matching framework on the simplex** (via spherical reparameterization) to rigorously model probabilistic disorder weights, coupled with a **Unified GNN** that captures complex multi-site geometric interactions.
2. **Benchmark and Performance:** We construct the first comprehensive benchmarks (**COD-SD/SPD**) and demonstrate that DMFlow significantly outperforms baselines (e.g., FlowMM, MatterGen) on both Crystal Structure Prediction (CSP) and De Novo Generation (DNG) tasks.

---
### **Summary of Discussions with Reviewers**
We have provided detailed, point-by-point responses to all four reviewers.

1. **Positive Engagement:** We are pleased to report that `Reviewer W2VS` has explicitly acknowledged the value of our new model extension and dataset contribution, subsequently **raising the score from 6 to 8 (prior to the information leak about two days)**.
2. **Pending Response:** `Reviewer XWs2` **initially provided a high score**, indicating a positive evaluation of the work, despite no further response during the rebuttal phase. `Reviewers y4YR and imvy` have also not yet replied, but **we believe our rebuttal has thoroughly addressed their concerns** regarding scope, baselines, and ablation studies with additional clarifications and new experiments. In light of their absence, we kindly ask that you take these rebuttal materials into consideration.

---
### **Summary of Key Concerns and Rebuttal Actions**
During the rebuttal, we provided detailed clarifications and conducted substantial new experiments to strengthen the paper:

1. **Extending to Higher-Order PD** (`Reviewers y4YR, imvy, W2VS`)

    + **Action:** To demonstrate generality, we significantly extended the framework to handle Higher-Order PD (up to 5 position splits). We generalized the representation to a **multi-channel formulation** and constructed a new benchmark (**COD-SHPD**).
    + **Result:** Empirical results confirm that DMFlow effectively captures complex spatial distributions (>97% coverage), proving its robustness beyond the binary setting.

2. **Baseline Comparisons and Performance Gains** (`Reviewers y4YR, imvy`)

    + **Action:** We clarified the non-trivial nature of adapting ordered-crystal baselines (e.g., FlowMM) to this task. Furthermore, we implemented and compared against a new SOTA baseline, **MatterGen**.
    + **Result:** DMFlow achieves chemically significant gains in geometric precision (e.g., **$\sim$26% RMSE reduction** on COD-SPD-20) and successfully breaks the MR-RMSE trade-off that limits adapted baselines.

3. **Rigorous Ablation Studies** (`Reviewers y4YR, XWs2, imvy`)

    + **Action:** We clarified that the comparison with simple flow matching was already present (Table 2) and added two new ablations: (1) comparison against a **Softmax-based Euclidean baseline**, and (2) a detailed evaluation of the **Voting/Discretization scheme**.
    + **Result:** These experiments conclusively validate the necessity of our intrinsic Riemannian flow matching (Simplex constraint) and the robustness of our ensemble discretization strategy.

4. **Clarifications on Task and Theory** (`Reviewers XWs2, imvy, W2VS`)

    + **Action:** We provided formal justifications for our experimental design, including the rationale for treating disorder weights as inputs in CSP, the data process of PD data for baselines and the adaptation of metrics for probabilistic structures.
    + **Result:** Specifically for `Reviewer imvy`, we provided a **mathematical derivation** proving how spherical reparameterization mitigates the numerical singularity of the Fisher-Rao metric. For `Reviewer W2VS`, we discussed the current limitations of thermodynamic stability evaluation for probabilistic structures, highlighting our work as a necessary step toward enabling such future assessments.

---

We hope this note assists you in reviewing our submission efficiently. Thank you again for your time and consideration.

Sincerely,

The Authors

---

### Meta-Review · Area_Chair_q78A · 2025-12-25

**Summary:**

This submission introduces a flow-matching generative framework aimed at modeling disordered crystals. The paper proposes a unified representation for ordered/SD/PD structures, Riemannian flow matching to maintain physically valid disorder weights, a symmetry-aware GNN, and a two-stage discretization scheme to convert continuous occupancies into discrete assignments. The authors curate and plan to release new disordered-crystal benchmarks from the Crystallography Open Database, which reviewers generally view as valuable.

The reviewers raise concerns about limited empirical evidence that the proposed modeling choices are necessary and robust, and about how much the contribution advances ML methodology beyond adapting existing ordered-crystal pipelines. Several reviewers question the generalizability of the discretization-heavy design, the significance of improvements, and the experimental protocol choices. The rebuttal adds clarifications and additional experiments (including higher-order PD and added baselines and ablations), the overall evidence remains insufficiently convincing for acceptance at ICLR given remaining questions about robustness, principled evaluation, and the necessity of multiple heuristics and thresholds in the generation pipeline.

**Reviewer Concerns:**

Addressed by the rebuttal

- Generality beyond binary PD. The authors added higher-order PD extension and a new benchmark/dataset (COD-SHPD).
- Baselines and ablations. The authors added MatterGen comparison and Euclidean normalization variants (L1/Softmax) to support the simplex/Riemannian design choice.
- Discretization analysis. The authors povided ablations of individual heuristics and a threshold sensitivity study; clarified CSP conditioning choices.

Still outstanding
- Heuristic dependence. The performance of the method still relies on multi-step discretization/voting with manual thresholds; attribution of gains and robustness across settings remain unclear.
- Evaluation mismatch. The model is probabilistic but is largely evaluated via discretized realizations and ordered-crystal-style metrics; no direct/proper evaluation of probabilistic disorder.
- CSP practicality. Treating disorder weights as observed in CSP sidesteps a harder, more realistic setting where disorder parameters must be inferred.
- Impact and novelty: Despite improvements, evidence for broadly compelling gains and a clear ML-method contribution beyond adaptation remains limited.

**Reviewer Scores:**

Reviewer y4YR (score 2): might moves up slightly (e.g., 2 to 3) given the added higher-order PD results and explicit Softmax baseline; still likely below threshold due to remaining concerns about marginal CSP gains and baseline breadth.

Reviewer XWs2 (score 6): likely stays around 6. Might increase to 7 if fully satisfied by the discretization ablation.

Reviewer imvy (score 4): likely moves modestly upward (e.g., 4 to 5) due to added baselines (MatterGen) and expanded ablations/derivation, but may remain unconvinced about ML novelty and evaluation appropriateness.

Reviewer W2VS (score 6): might increases by 1 due to model extension and improved clarity; but unlikely vouching for the work.

---

### Decision · Program_Chairs · 2026-01-26

Reject